# Adult Neurogenesis: A Story Ranging from Controversial New Neurogenic Areas and Human Adult Neurogenesis to Molecular Regulation

**DOI:** 10.3390/ijms222111489

**Published:** 2021-10-25

**Authors:** Perla Leal-Galicia, María Elena Chávez-Hernández, Florencia Mata, Jesús Mata-Luévanos, Luis Miguel Rodríguez-Serrano, Alejandro Tapia-de-Jesús, Mario Humberto Buenrostro-Jáuregui

**Affiliations:** 1Laboratorio de Neurociencias, Departamento de Psicología, Universidad Iberoamericana Ciudad de México, Ciudad de México 01219, Mexico; mariele_chavez@yahoo.com (M.E.C.-H.); maria.florencia@ibero.mx (F.M.); jarmando.luevanos@gmail.com (J.M.-L.); cosmonauta84@yahoo.com.mx (L.M.R.-S.); tapia.neuropsic@gmail.com (A.T.-d.-J.); 2Laboratorio de Neurobiología de la Alimentación, Facultad de Estudios Superiores Iztacala, Universidad Nacional Autónoma de México, Tlalnepantla 54090, Mexico

**Keywords:** adult neurogenesis, subventricular zone, striatum, hippocampus, habenula, cerebellum, substantia nigra, hypothalamus, neurotrophin, microRNA

## Abstract

The generation of new neurons in the adult brain is a currently accepted phenomenon. Over the past few decades, the subventricular zone and the hippocampal dentate gyrus have been described as the two main neurogenic niches. Neurogenic niches generate new neurons through an asymmetric division process involving several developmental steps. This process occurs throughout life in several species, including humans. These new neurons possess unique properties that contribute to the local circuitry. Despite several efforts, no other neurogenic zones have been observed in many years; the lack of observation is probably due to technical issues. However, in recent years, more brain niches have been described, once again breaking the current paradigms. Currently, a debate in the scientific community about new neurogenic areas of the brain, namely, human adult neurogenesis, is ongoing. Thus, several open questions regarding new neurogenic niches, as well as this phenomenon in adult humans, their functional relevance, and their mechanisms, remain to be answered. In this review, we discuss the literature and provide a compressive overview of the known neurogenic zones, traditional zones, and newly described zones. Additionally, we will review the regulatory roles of some molecular mechanisms, such as miRNAs, neurotrophic factors, and neurotrophins. We also join the debate on human adult neurogenesis, and we will identify similarities and differences in the literature and summarize the knowledge regarding these interesting topics.

## 1. Introduction

The adult brain possesses the capacity to produce new neurons throughout life. This process, called neurogenesis, comprises a series of sequential events that are necessary for the generation of new neural cells [1]. The generation of new neurons has been controversial since its very first description in 1965 by Altman and Das because this concept was incongruous with the traditional paradigm of neuroscience [2]. Since then, an extensive amount of data has been generated regarding how and where these neurons are generated. To date, new neurons have been reported to develop postnatally in several species, including humans [3,4,5]. The generation of these neurons arises from populations of precursor cells that reside in restricted areas of the brain and generate new neurons throughout life [6]. New neurons have electrophysiological properties that allow them to integrate into the existing circuitry in early stages [7] and functionally participate in tasks related to memory and learning [8]. Additionally, the number of neuronal generations decreases with natural aging [9,10]. Although new neurons generated in the adult brain differ in their functional specialization, some mechanisms involved in the regulation of the phenomenon are shared. We will first begin by defining some important concepts for naive readers. The term neural stem cells (NSCs) refers to primary progenitor cells at different developmental stages that initiate lineages that lead to the formation of differentiated neurons or glial cells [11]. A neurogenic niche is a remnant of the embryonic germinal layer region with a special microenvironment that preserves NSCs, allowing and regulating NSC activity [11] and culminating in adult neurogenesis. The niches or adult neurogenic sites share features, mainly in the more primitive precursor cells, which are described as radial glia-like cells. These populations develop into different subsets of cells during a process consisting of several steps until they become mature, fully integrated cells. Spontaneous or constitutive adult neurogenesis was once considered a rare phenomenon in mammals. Over the decades, the major regions studied and considered neurogenic have been restricted to only two adult neurogenic sites: the forebrain subventricular zone (SVZ) [12,13] and the subgranular zone (SGZ) of the hippocampal dentate gyrus (DG) [6,14]. The vast majority of mammalian brain regions (not including neurogenic sites) are considered nonneurogenic. However, several subcortical areas, such as the hypothalamus, substantia nigra, striatum, amygdala, habenula, and cerebellum, have been reported as neurogenic in recent decades (see Figure 1). Interestingly, neurogenic zones give rise to fully mature neurons with different functional specializations related to the function of the area in which they reside. The SVZ gives rise to dopaminergic cells as well as γ-aminobutyric acidergic (GABAergic) and glutamatergic cells that migrate from the rostral migratory stream and integrate into the olfactory bulb [15,16,17]. Notably, the hippocampal DG generates cells that become granular cells and participate in memory, learning, and cognitive flexibility [8,10,18,19,20,21,22]. New cells in the hypothalamus specialize in energy balance and other homeostatic mechanisms [23,24,25]. Neurogenesis in the substantia nigra has been linked to the dopaminergic system. However, open questions regarding its functional implications in local circuitry still persist [26]. Few new neurons are created in the striatum in a constitutive manner. However, in response to lesions, neurogenesis in this area can increase, producing fully integrated neurons [27]. Newly generated cells in the habenula are associated with the circadian cycle [28] and stress response regulation of the antidepressant effect of fluoxetine [29]. Researchers hypothesized that new granule cells and interneurons in the cerebellum potentially play an important role in learning/adapting motor skills to the environmental cues the animal encounters during its life [30].

The phenomenon of adult neurogenesis is not free of controversy, particularly in relation to two fundamental issues: adult neurogenesis in humans (mainly at advanced ages), and the true presence of neurogenic niches in new brain regions. In this manuscript, we review most neurogenic zones, from the traditional constitutive zones to the less described zones, and their functional relevance to provide the reader with comprehensive knowledge of what is currently known about adult neurogenesis phenomena. We include an approach to this phenomenon that includes other components that are not usually considered, such as miRNAs and neurotrophins. Additionally, we will discuss the criticism of some of the new neurogenic sites, particularly the habenula and the cerebellum. We will also join the debate on human adult neurogenesis.

## 2. The Traditional Neurogenic Zones: SVZ and SGZ

### 2.1. Adult Neurogenesis in the SVZ: Mechanism and Possible Functional Implications

The SVZ of the lateral ventricle is one of the two areas of the brain where adult neurogenesis occurs [31]; more specifically, new neurons are generated from stem/progenitor cells in this region [16]. The NSCs that populate this neurogenic niche are known as type B1; they resemble astrocytes and then differentiate into neurons that populate the olfactory bulb (OB) [16,32]. Neurogenesis in the SVZ continuously provides new GABA- and dopamine (DA)-containing interneurons in the OB [15]. NSCs also differentiate into astrocytes and oligodendrocytes throughout a person’s life [32]. These type B1 cells that line the lateral ventricles express glial fibrillary acidic protein (GFAP, an intermediate filament protein that provides support and strength to cells), glutamate aspartate transporter, and brain lipid-binding protein [16]. Importantly, NSCs coexist in one of two stages in the adult SVZ: quiescent (qNSCs) or actively dividing (aNSCs) [33].

Differentiation within the SVZ is divided into three stages.

When type B1 cells are activated, they either express nestin and divide asymmetrically for self-renewal, or give rise to achaete-scute homolog 1 and distal-less homeobox 2-expressing C cells [16].These type C cells then divide symmetrically two to three times and subsequently differentiate into type A cells (neuroblasts), which represent the final stage of differentiation within the SVZ [16,33].These type A cells then divide one or two times and migrate through the rostral migratory stream (RMS) toward the OB [16,33].

Neurogenesis in the SVZ occurs throughout adulthood in the mammalian brain, and neurogenesis in this region is known to play an important role in the development of optimal olfactory circuitry (see Table 1). Constant granule cell regeneration and replacement allow mammals to respond to new environmental stimuli and reinforce particular odorant representations that are more pervasive in the environment. Additionally, since steroid hormones influence SVZ/OB neurogenesis, these newly generated neurons probably have a role in sexual function [16].

Several intrinsic and extrinsic factors, such as growth factors, signaling pathways, neurotransmitters, and genes, play a role in the modulation of neurogenesis in the adult SVZ [16,31]. Furthermore, neurogenesis in the SVZ can occur as a result of neurological diseases, such as stroke or seizures [42], or physical exercise [43,44,45]; moreover, an enriched environment [46,47,48] modulates neurogenesis in the SVZ after an event of this kind. Regarding neurotransmitters, serotonin (5HT) and serotoninergic transmission play a critical role in the initial stages of SVZ cell proliferation [16], increasing proliferation and neurogenesis; in particular, 5HT increases the proliferation of B1 cells through the activation of 5HT2C receptors [40]. Additionally, cholinergic neurons in the SVZ (which expresses choline acetyltransferase) are postulated to regulate neuroblast proliferation through the activation of fibroblast growth factor receptor (FGFR)-mediated signaling [16,49]. Through optogenetic inhibition and stimulation, cholinergic neurons have been shown to control neurogenic proliferation in vivo [49]. Additionally, GABAergic neurons inhibit cell proliferation and neuronal differentiation in this neurogenic niche [16]; for example, while GFAP-expressing cells generate neuroblasts, GABA released from neuroblasts participates in a feedback mechanism that controls the proliferation of GFAP-expressing progenitors through the activation of GABA_A_ receptors [41]. The main growth factors involved in SVZ neurogenesis are expressed by astrocytes and provide signals to NSCs in the SVZ, including mitogen fibroblast growth factor 2 (FGF-2) and epidermal growth factor [16], promoting astrocyte hypertrophic morphology and proliferation [39].

Some additional genes influence neurogenesis in the SVZ. For example, the KAT6B (Lysine Acetyltransferase 6B) gene is expressed at high levels in this neurogenic niche and plays an important role in adult neurogenesis [37]. Similarly, KAT6B-deficient mice have reduced numbers of NSCs and migrating neuroblasts in the RMS [31]. Additionally, the Btg1 gene is known to control the proliferation of stem/progenitor cells in the SVZ and is associated with the maintenance and self-renewal of stem cells in this niche [38,50].

As mentioned above, physical exercise exerts neurogenic effects, but these effects are not normally as strong on the SVZ as on other neurogenic niches. However, studies have shown that exercise stimulates neurogenesis in this niche in response to factors that alter this process [43,44,51]. A study performed by Lee et al. sought to determine whether corticosterone suppressed cell proliferation in the SVZ and whether this effect was reversed by voluntary exercise [45]. The authors found that rats treated with chronic corticosterone (4 weeks) exhibited decreased cell proliferation (indicated by immunodetection of the thymidine analog 5-bromo-2′-deoxyuridine (BrdU), which is a standard technique used to visualize these newly generated cells) [52] in the SVZ, whereas rats that were only subjected to voluntary exercise exhibited increased cell proliferation in this neurogenic niche. Furthermore, rats receiving both treatments (chronic corticosterone and simultaneous wheel running) had approximately the same number of BrdU-labeled cells as those in the control group; additionally, these BrdU-labeled cells expressed doublecortin (DCX), a migrating neuroblast marker. Notably, the percentage of BrdU-labeled cells expressing DCX was decreased by treatment with corticosterone, while exercise increased this percentage. Overall, wheel-running exercise alleviates the effects of corticosterone in terms of the suppression of cell differentiation [45]. Another study by Nicolis di Robilant et al. focused on the effects of three different paradigms of voluntary exercise on SVZ neurogenesis in p21 knockout (KO) mice that were studied at two different stages of development: 2 and 12 months of age [43]. Physical activity activated and expanded NSCs and enhanced SVZ neurogenesis. Furthermore, the authors concluded that 12 days of running was a sufficient amount of exercise to increase the number of new neurons that were functionally active in the OB. Functionally active neurons in this region are associated with improved olfactory performance that strictly depends on adult neurogenesis in the SVZ, as measured by the odor detection threshold and short-term olfactory memory tests. The results of this study show that NSCs in the adult SVZ of p21 KO mice have a high neurogenic potential that is triggered by physical activity and potentially results in long-term consequences in olfactory-related behaviors [43]. In another study by Mastrorilli et al., 12 days of voluntary running were associated with fully restored neurogenesis in the SVZ of Btg1 KO mice [44]. In their study, 2-month-old Btg1 KO mice were subjected to 5 or 12 days of voluntary running; it was found that running specifically increased neurogenesis in the SVZ of these mice. The number of stem cells (B cells) and neuroblasts (C cells) in the KO mice increased significantly compared with the numbers observed in the control mice [44]. Additionally, a study by Farioli-Vecchioli et al. showed that 12 days of physical exercise (running) fully reversed deficient adult neurogenesis within the hippocampus and SVZ of mice lacking the Btg1 gene [53]. Using GFAP as a marker to specifically detect type B NSCs and DCX to detect type A progenitor cells, the authors showed that running fully reactivated the postnatal hyperproliferation of newly generated subventricular neurons in Btg1 KO mice [53].

As mentioned earlier, neurogenesis in the SVZ also occurs as a result of neurological diseases, such as stroke. A study by Palma-Tortosa et al. showed longitudinal changes in neurogenesis in the SVZ of a mouse model of cortical ischemia [54]. The authors observed a triphasic effect of stroke on the number of proliferating cells in the SVZ: first, an early acute reduction in proliferation occurred on poststroke day 1, followed by a slow increase in proliferation from days 2 to 7 after cortical ischemia, reaching a maximum on poststroke day 14. Finally, a reduction in the number of proliferating cells was observed 28 days after cortical ischemia. Additionally, the authors showed that the early acute decrease in the number of proliferating cells at the SVZ might have occurred due to increased neuroblast migration through the physiological RMS after stroke (on day 1), which supported the hypothesis that this increased migration toward the OB explains the decrease in the number of simultaneously proliferating cells in the SVZ [54].

Exposure to an enriched environment (EE) after stroke has been shown to exert neuroprotective effects [46,47,48]. For instance, a study by Zhang et al. (2018) showed that mice housed in an EE 2 days after middle cerebral artery occlusion (MCAO), an animal model of ischemia, exhibited 51% more BrdU/DCX-labeled cells in the SVZ than sham mice (mice that underwent surgery without MCAO) housed in an EE at 28 days after injury [47]. Furthermore, the number of BrdU/DCX cells in the ischemic SVZ was significantly increased by 40% and 38%, respectively, in MCAO model mice housed in an EE compared to MCAO model mice housed in a standard environment 28 days after injury; thus, an EE exerts a therapeutic effect on stroke, increasing poststroke neurogenesis in the SVZ [47].

Another study by Tang et al. showed that the effects of EE exposure are not limited to the acute phase after stroke [46]. In their study, rats were exposed to an EE for three weeks, starting 5 days after MCAO. The authors showed that exposure to an EE in the delayed phase (5 days later) significantly ameliorated ischemia-induced impairments in the cognitive performance in specific tests (measured by the Morris water maze) of MCAO model mice compared to mice in the control group. Additionally, the migration of SVZ-derived cells toward the ischemic striatum was increased [46].

### 2.2. Adult Neurogenesis in the SGZ of the Hippocampus: Mechanism and Possible Functional Implications

Adult hippocampal neurogenesis (AHN) is now the prevailing dogma in neuroscience, at least for mammals other than primates [55]. AHN may be an evolutionarily novel system that develops to increase structural and functional plasticity in the hippocampus [56]. In this regard, the hippocampus receives a large number of sensory inputs unidirectionally from neocortical regions, which regulates the hippocampal formation. Additionally, the DG and CA3 regions of the hippocampus are implicated in spatial memory function and the capacity for pattern separation, and are associated with an organism’s ability to learn new information [57]. Additionally, the DG, a part of the hippocampal formation, has important functions in learning, memory, and adult neurogenesis (Table 2 shows a summary of the material presented here). Additionally, the DG of mammals has characteristics that distinguish it from its counterparts in other vertebrate species. For example, the DG is larger, convoluted, and has non-periventricular neurogenesis [58]. Adult neurogenesis declines during physiological aging in mammals. In this regard, both< intrinsic and environmental factors influence mammalian neurogenesis, including in aged animals [59]. Hollands et al. suggest that deficits in adult neurogenesis may contribute to cognitive impairments, tau hyperphosphorylation in new neurons, and compromised hippocampal circuitry in subjects with Alzheimer’s disease [60].

The SGZ of the hippocampal DG is one of the neurogenic niches of the adult brain, where the generation of new neurons persists throughout adulthood [10,56,68,69]. In addition, the DG is primarily composed of granule cells that are located in the molecular and granule cell layers. Located between the granule cell layers and CA3 region, the hilus contains granule cell axons that are labeled mossy fibers, mossy cells, and a variety of GABAergic interneurons [70]. This thin band between the granule cell layer and the hilus provides a unique microenvironment for an adult NSC population. The permissive milieu of the SGZ allows NSC proliferation while promoting the specification and differentiation of DG neurons [71]. In the SGZ, newly generated neurons migrate to the granule cell layer, where they mature and develop dendritic branches [72]. These newly generated neurons integrate into the existing local DG network, where they acquire many of the long-range connections typical of mature granule cells and synapse locally upon inhibitory interneurons and mossy cells [73]. In addition, DG cells receive excitatory inputs from diverse cortical and subcortical circuits, and other hippocampal subregions are well positioned to relay signals to NSCs and immature adult-born DG cells [74]. The hippocampal niche generates new cells that give rise to neurons during a step in the developmental process that involves the asymmetric division of subsets of neural precursor cells. These subsets include type-1 cells, which have radial glia-like features, as well as type-2a cells and type-2b cells, which already regulate neuronal cell fate and the process that culminates in the development of a new neuron [6]. In this regard, Kozareva et al. proposed five principal developmental stages of AHN, starting with radial glia-like cells and continuing through development to progenitor cells, neuroblast cells, immature neurons, and finally, mature neurons as granular cells [75]. Furthermore, the structural organization of the DG neurogenic niche suggests extensive cross-talk between niche cell types and adult-born DG cells and between position NSCs and adult-born DG cells; hence, adult-born DG cells serve as integrators of diverse signals from the niche. Additionally, AHN is a highly regulated process that leads from a quiescent state to neuronal differentiation [76] and is also an activity-dependent process that enables the formation of a new neural network [77]. In this regard, AHN initially produces rapidly proliferating precursor cells, followed by differentiation along the neurogenic trajectory, as the newborn cells exit the cell cycle and become immature neurons, and finally, the neurons survive and mature to become fully functional granule cells that are then integrated into the neural network [78].

AHN is proposed as a continuous developmental process that helps maintain a highly plastic network to add immature neurons into the pre-existing neural network such that mature neurons are ultimately integrated functionally and structurally [79]. Several intrinsic and extrinsic factors positively and negatively influence the formation of new neurons during this process. During AHN, a combination of extrinsic and intrinsic cues interact in regional intrinsic programs to increase neuronal diversity [80]. For example, extrinsic factors that improve AHN include environmental enrichment [10,81,82], aerobic exercise [81,83,84,85], diet—which can promote AHN [86,87] or obstruct AHN [88]—and sexual behavior [89]. Although other extrinsic factors, such as stress and disease, exert a negative effect on AHN, aging is the key factor contributing to the decrease in natural AHN [9,90]. On the other hand, intrinsic factors, such as multilayered regulatory networks, consist of connections between multiple complex transcription factors, epigenetic control, noncoding RNAs, signaling, and metabolic pathways [91]. Additionally, canonical Wnt signaling is fundamental for the proper development of the cortex and hippocampus during embryonic development. In addition to promoting self-renewal and maintaining neural progenitors during early neurogenesis, canonical Wnt signaling induces the differentiation of intermediate progenitors during mid and late neurogenesis [71]. Furthermore, several studies indicate that Wnt proteins released by hippocampal astrocytes [61,92] and progenitor cells are crucial components of the SGZ niche [93]. Additionally, some ligands of Wnt promote AHN through a noncanonical pathway, such as Wnt5a. AHN induction by these pathways was shown to be an important neurogenic factor involved in the process of neuronal differentiation, as well as in the morphological development of newly generated neurons in the hippocampus through calcium/calmodulin-dependent protein kinase II (CaMKII); new neurons generated through this pathway induce neuronal differentiation and promote the development of dendrites extending from new neurons via the Wnt/JNK and Wnt/CaMKII pathways [94].

On the other hand, sonic hedgehog (Shh) is a multifunctional signaling protein that participates in cell formation, proliferation, and survival during embryogenesis [62,95]. Furthermore, the Shh protein was recently shown to regulate AHN [96]. Ablation of the Shh protein in the adult brain induces increases in the number of neural precursors, proliferation, and migration during AHN [62]. Impaired Shh signaling may contribute to the pathogenesis of several developmental disorders, some of which affect the hippocampus [95].

Additionally, Anacker and Hen suggest that young neurons may contribute to the circuitry that regulates information processing in AHN by integrating new information [97]. Furthermore, Kirschen et al. show that a mature DG circuit may not participate directly in the generation of new memories per se but rather modulate them [98]. Additionally, researchers have started to determine how new neurons formed through AHN participate in memory processes and how these new neurons in the adult brain contribute to memory throughout the lifespan [99]. For example, Cope et al. suggested that AHN is essential to maintain social memory but not for its acquisition or retrieval over a short period [100]. In summary, these new neurons entering the adult brain are part of the neural plasticity that results from the continuous integration of newly born neurons into the adult hippocampus. Furthermore, AHN was shown to be an activity-dependent process that generates changes in the plastic structure and function of the adult brain rather than a simple continuation of embryonic and fetal neurogenesis [101].

## 3. Beyond the SVZ and DG: New Adult Neurogenesis Zones

The periventricular region is a common neurogenic area in all vertebrates, although as we move away from the paraventricular region, these distal areas display many differences between vertebrates [102]. The presence of multiple neurogenic niches in adulthood has been reported in different animals, ranging from fish [28] to mammals [23], including humans [4,103].

Below, we will present those brain structures for which evidence of adult neurogenesis has been obtained, either constitutively or after experimental manipulation with drugs or genetic tools. We will begin with the brain structures with the most evidence (the hypothalamus, substantia nigra, striatum, and amygdala). Finally, we will end with two of the most criticized and controversial zones, for which there is limited but promising evidence (the habenula and cerebellum).

### 3.1. Adult Neurogenesis in the Hypothalamus: Mechanism and Possible Functional Implications

Hypothalamic neurogenesis occurs constitutively in the adult brain [104]. The hypothalamic neurogenic niche is located in the subependymal zone of the third ventricle (the hypothalamic ventricular zone, HVZ); however, unlike the SVZ and SGZ, cell proliferation in this region is not restricted to cell layers. The new neurons are spread over the hypothalamic parenchyma [24,105]. The population of precursor cells in the hypothalamus is known as tanycytes. These cells are recognized as radial glial-like cells in the circumventricular organs. Currently, these cells are identified as hypothalamic radial glial cells [106]. The neurogenic niche in the hypothalamus is located in the median eminence, known as the proliferative domain of the hypothalamic proliferative zone (HPZ). This area was established based on the criteria of the position of the tanycytes and co-labeling with precursor cell markers [107]. The hypothalamic niche is composed of several subsets of cells that are located in different positions along the third ventricle. These cell populations have been named tanycytes α and β. These cells are categorized as either tanycyte α1 cells, which are located in the ventromedial nuclei in the third ventricle, or tanycyte α2 cells, which are located by arcuate nuclei. These two populations extend long processes toward blood vessels and connect with the hypothalamic network and glial cells. Tanycyte β cells are also divided into two subpopulations. These subpopulations are located in two regions: tanycyte β1 resides in the lateral part of the infundibular recess, whereas tanycyte β2 forms the HPZ at the bottom of the third ventricle in the median eminence [107,108,109]. The neurogenic activity of the proliferative niche was observed up to postnatal day 75, confirming that it represents an adult neurogenesis phenomenon [107]. The cell fate of newly generated neurons was shown using a multiple labeling approach with BrdU and nestin/GFAP immunohistochemistry and co-labeling with GFP-recombinant adenoviral infection (vGFP), and some of the tanycytes in the area were shown to have precursor cell features. Additionally, the group labeled with the adenoviral tracer migrates from the third ventricle to the parenchyma of the hypothalamus, where the cells integrate into the existing network and become active [110]. The adult hypothalamic NSC niche has the peculiarity of being distributed in two zones: the hypothalamic ventricular zone (HVZ, located in the lateral walls of the third ventricle, at the level of paraventricular and arcuate nuclei) and the HPZ. Immunofluorescence staining has revealed that tanycytes express proteins associated with neural precursor cell features, such as the intermediate filament protein nestin [111,112]; vimentin, a marker of precursor cells [113]; and DCX, a marker of young neurons [114]. The newly generated neurons differentiate into neurons of the hypothalamic network expressing markers such as Agouti-related peptide, an orexigenic factor observed in neurons within the arcuate nucleus [23]. Additionally, components of the Notch pathway were observed in these cell populations [115]. Table 3 provides a summary of the material presented here.

Cells in the hypothalamic neurogenic zone express markers of neural stem and progenitor genes, such as Sox9, Sox2, Notch 1 and 2, Hes 1 and 5, CD63, FZD5, Dirc, NTrk-2T1, and Thrsp [107,108,115]. Additionally, co-labeling of BrdU-positive cells in the hypothalamic proliferative niche and Hu+, a widely known marker of progenitor cells, was reported [107].

The newly generated cells in the hypothalamus are functionally related to the energy balance and various hypothalamic homeostatic mechanisms [23,24,25]. Lee et al. showed that animals fed a high-fat diet for 30 days beginning on postnatal day 45 exhibited a significant increase in the number of Hu+/BrdU+ cells located at the base of the third ventricle of the median eminence, indicating an increase in adult hypothalamic neurogenesis in adult animals [107]. This outcome is not observed in younger animals, which might also suggest an aging-related effect due to fat intake [107].

Neurogenesis in this region is stimulated by insulin-like growth factor I (IGF-I). An experiment conducted with microdoses of IGF-I delivered with a cannula implanted into the right lateral cerebral ventricle showed that local neurogenesis increased proliferation; additionally, these newly generated cells survived for longer time periods and regulated cell fate [104].

### 3.2. Adult Neurogenesis in the Substantia Nigra: Mechanism and Possible Functional Implications

The substantia nigra (SN) is a structure located in the mesencephalon. It is anatomically and functionally divided into two parts: the pars compacta and the pars reticulata (for a further review, see [117]). It receives various inputs from different areas, such as the subthalamic nucleus, the amygdala, the cortex, the laterodorsal tegmental nucleus, the habenula, and the pedunculopontine tegmental nucleus, but its principal sources of input are the caudate–putamen complex and the pallidum (see [118]). Although the SN has various functions, its most relevant role is in movement (motor planning and eye movement). The pars compacta of the substantia nigra (SNpc) is a midbrain area that contains DA-producing neurons, which are typically lost in individuals with Parkinson’s disease. Located in the caudal and dorsal (posterior and inferior) part of the SN, the SNpc contains melanized neurons, giving the color after which it is named.

The SN has been described as playing an important role in producing and releasing DA in the motor and reward systems, but recently, it has been implicated in the regulation of sleep (for an extensive review, see [119]). The SN pars reticulata is located in the rostral and ventral (superior and anterior) parts of the SN. It has a lamellar arrangement (onion-like), forming a set of nuclei that connect to the thalamus, the superior colliculus, and the tegmental area [120]. It is divided into medial and lateral portions; mediates functions, such as the sensory-motor system; and has an important, yet novel, role in regulating and maintaining sleep [121]. Regarding adult neurogenesis in this area, Zhao et al. provided evidence suggesting neurogenesis in the SNpc [26]. The authors administered BrdU (i.p., orally and i.c.v.) and traced new neurons in the SNpc with immunostaining. Moreover, the authors found that BrdU administered for 2 days did not result in immunostaining, but new neurons were observed after 10 or 21 days of BrdU administration. This finding might explain why the immunostaining was due to the generation of new neurons and not to DNA repair. Furthermore, stem cells lining the ventricles were labeled with BrdU. These stem cells can become new tyrosine hydroxylase (TH)-positive neurons in the SNpc, as shown by Zhao et al. when they used BrdU and 1,1′-dioctadecyl-3,3,3′,3′-tetramethylindocarbocyanine perchlorate (DiI) to mark and trace new neurons from the ventricles to the SNpc [26]. This neurogenesis in the SNpc occurs at a lower level than that in the DG of the hippocampus but is still important to note, since neurogenesis outside key areas, such as the hippocampus and the OB, is a relatively new concept. This so-called small-scale neurogenesis might be a homeostatic mechanism since researchers have hypothesized that the number of neurons in the SNpc decreases with aging. Nevertheless, no further proof of small-scale neurogenesis as a mechanism of homeostasis has been obtained. However, after lesions are generated in the SNpc (a model that tries to imitate Parkinson’s disease), neurogenesis in this area is enhanced. Zhao et al. [26] administered a peripheral dose of 1-methyl-4phenyl-1,2,3,6-tetrahydropyridine (MTPT), a known toxic agent that destroys half of the nigral dopaminergic nerve cell population. After MTPT administration, animals received daily doses of BrdU and were then sacrificed 2, 10, or 21 days later before an immunostaining assay was conducted. The authors found no changes in the hippocampus; however, changes in the SNpc were observed beginning on day 10 but were more robust on day 21. The lesions enhanced neurogenesis. This result explains how the toxic agent (specific to dopaminergic neurons in the SNpc) caused changes in the SNpc but not in the hippocampus through lesion-led neurogenesis [26]. Despite the results described by Zhao and colleagues, another study was carried out in which 1-methyl-4-phenylpyridinium ion (MPP), a proposed analog of MTPT, was administered; the administration of MTPT resulted in the same severity of the lesion and behavioral and cellular response. After 7 days of i.c.v. MPP administration, an increase in the number of BrdU-positive cells in the subgranular zone of the DG of the hippocampus was observed; nevertheless, a reduction in the number of BrdU-positive cells in the SN was detected [122]. The contradictory results were due to the use of different substances, probably with different effects and different mechanisms for modulating dopaminergic neurons. The last statement may be supported by the work of another research group in which the researchers lesioned mice with MPTP and found a depletion of TH-positive neurons (suggesting damage to the SN). Afterward, they labeled cells with BrdU to detect neurogenesis and found that new cells were being generated without a lesion, but this effect was augmented after a lesion was induced [123]. New cells were identified by Shan et al., including TH-positive neurons, suggesting that neurogenesis in the SN results in an increase in the number of dopaminergic neurons [123].

Although neurogenesis occurs, apoptosis processes may increase the difficulty of perceiving neurogenesis since researchers studied TH+ neurons (the rate-limiting enzyme for DA synthesis) over the life span of a mouse. The researchers found that this number remained stable, without any significant difference, despite indicators of apoptotic processes, such as shrunken TH+ neurons and condensed terminal deoxynucleotidyl transferase-mediated dUTP-biotin nick end labeling (TUNEL; a known marker of apoptosis or cell death)-positive nuclei [26].

More studies are needed to elucidate the relationship between these two toxins and potential SN neurogenesis. In particular, although these studies drew different conclusions, Park et al. did not find differences in the number of BrdU-positive cells between a control group and an experimental group treated with MTPT [124].

Consistent with previous work, an experiment was performed in which positive immunostaining was detected in the SN following the administration of BrdU. As a supplement to this investigation, Lie et al. detected new cells as early as 2 h after BrdU administration, and even more new cells were observed after 3 days of BrdU injections [125]. These new cells were present in doublets, suggesting that a cellular division process occurred locally. Lie et al. performed TH staining to determine if the cells were dopaminergic neurons and to obtain a better understanding of the nature of these new cells, but did not detect positively labeled cells, excluding the possibility that these new cells were neurons [125]. After performing GFAP labeling and observing a positive reaction, these cells were considered glia. Taking this information into account, the SN has the potential for neurogenesis, although the process of generating new cells does not tend to promote differentiation into neurons.

Another study did not observe TH-positive neurons after MTPT administration, but due to labeling with green fluorescent protein (GFP) and with a glial marker, the researchers found that the newly generated cells are most likely glia (especially microglia) [126]. Table 4 shows a summary of the material presented here.

In an elegant study, microinfusions of 7-hydroxy-N,N-di-n-propyl-2-aminotetralin (7-OH-DPAT), a preferential agonist of the DA D3 receptor, or vehicle (saline) were administered into the ventral third ventricle of female Sprague-Dawley rats. In addition to this treatment, rats received a daily i.p. injection of BrdU (50 mg/kg). Saline treatment resulted in a small number of BrdU-positive cells in the area surrounding the third ventricle and in the SN, but surprisingly, a greater (compared to the saline) number of BrdU-positive cells was observed in the same areas in the experimental group treated with chronic infusion of the D3 agonist [127]. Van Kampen and Robertson employed distinct markers to elucidate the cell identity and found that BrdU-positive cells colocalized with TH (a known dopaminergic marker) [127]. Hence, treatment with the agonist not only promoted neurogenesis but also promoted neuronal differentiation into the DA phenotype. The mechanism underlying this particular type of neurogenesis remains to be elucidated and could reveal potential new therapeutic targets.

In addition, other molecules and processes are related to neurogenesis in the SN, such as gene mutations. In humans, an early onset of Parkinson’s disease related to the expression of an autosomal recessive mutation in (D221Y) PLA2G6 was reported [130]. Chiu et al. established a rodent model with a knock-in of this mutation that replicated early-onset Parkinson’s disease in mice [130]. The mitochondrial cristae of three groups were measured and compared: animals homozygous for the mutation, heterozygous animals, and wild-type animals. The mitochondria of wild-type and heterozygous mice were similar compared to those of homozygous mice, which had smaller mitochondria and disrupted cristae. This result might explain how some gene alterations may affect some molecular systems, altering functionality, morphology, and neurogenesis. These results might also be relevant to the clinical area since the transplantation of potential DAergic cells could be performed. However, this process has not yet had a great effect on motor activity. In a study where transplanted cells developed into DA cells in the SNpc, the SNpc of the host expressed nestin and DCX (known markers for neurogenesis) after transplantation. Moreover, when the authors transplanted cells into the SNpc damaged by 6-hydroxydopamine (6-OHDA), they detected TH+ cells [131]. Based on these results, the SN not only has the ability to produce new neurons on a regular basis but also the potential to enhance this system after it has been lesioned or damaged.

### 3.3. Adult Neurogenesis in the Striatum: Mechanism and Possible Functional Implications

The striatum is a forebrain structure that receives GABAergic and glutamatergic inputs from different sources and coordinates aspects of motor behavior and responses to rewarding and aversive stimuli. The striatum is divided into the ventral striatum (the nucleus accumbens and olfactory tubercle) and the dorsal striatum (the caudate nucleus and putamen) [132].

Adult neurogenesis in the striatum is limited under normal physiological conditions but can be induced by different procedures, for example, in response to different pathological stimuli, such as stroke/ischemia or injury, and pharmacological stimuli, such as an infusion of some growth factors and neurotrophins, in animals and human models. Several studies have shown two possible sources of newly generated neurons in the striatum. The first are precursors from the SVZ because neuroblasts born in this region have been proposed to migrate toward the striatum [27,133,134]. The second are local neuronal precursors in the striatal parenchyma because it contains progenitor cells that are activated and become neurogenic when stimulated with neurotrophic factors [135]. This adult neurogenesis in the striatum might be a compensatory mechanism by which the damaged adult brain tries to repair itself.

Sufficient experimental evidence supports the hypothesis that stroke-mediated damage to the brains of adult rats, rabbits, monkeys, and humans induces neurogenesis in the striatum. The initial findings were reported by Arvidsson and colleagues (2002), who showed that new neurons are generated in the adult striatum of rats after stroke caused by transient MCAO. New neurons are generated as precursors in the SVZ and migrate toward the damaged area of the striatum, where they differentiate into medium-sized GABAergic spiny interneurons in the striatum. These new neurons expressed markers for proliferation (BrdU), as survival (DCX, Hu, Mies homeobox 2 (Meis2) and Pbx) and mature (NeuN and DARPP-32) striatal spiny neurons [136]. In another study, newly formed neurons resided in the nucleus accumbens and dorsomedial striatum, where these neurons also expressed the markers BrdU, DCX, NeuN, collapsin mediator response protein 4 (CRMP4), glutamic acid decarboxylase 67 (GAD-67) and calretinin (CR) [137]. Notably, a study showed that newborn GABAergic neurons are electrically active and capable of firing action potentials and receiving excitatory and inhibitory inputs, suggesting that these neurons could become functionally incorporated into the neuronal networks in the brains of adult rats after stroke [138].

Interestingly, another study used transgenic mice carrying a green fluorescent protein (GFP) gene that were injected with a Cre-encoding recombinant adenovirus into the lateral ventricle, which specifically labels SVZ cells and their progeny; the authors observed GFP-labeled cells that expressed DCX and NeuN in the striatum after stroke [139]. In addition, the infusion of epidermal growth factors (EGF) and fibroblast growth factor-2 (FGF-2) in the lateral ventricle of adult rats also increased BrdU and NeuN levels after ischemia [140]. Moreover, another study combined two experimental procedures in vivo, namely, an infusion of transforming growth factor alpha (TGF-α) was administered into the striatum of adult rats and the unilateral lesion generated by 6-hydroxydopamine (6-OHDA) in the dopaminergic neurons of the substantia nigra. These two treatments generated new neurons in the SVZ that expressed BrdU and NeuN [27]. Similarly, a study using a quinolinic acid (QA) lesion model in adult rats revealed that QA lesion-induced striatal cell loss generated an increase in the number of BrdU-labeled cells in the SVZ and led to the migration of neurons to the lesioned striatum, where these cells expressed DCX [141].

Notably, studies with adult macaque monkey brains showed an increase in the proliferation of BrdU-labeled cells in the SVZ after ischemia and the restriction of these cells from migrating toward the olfactory bulb but not toward the striatum [142]. In contrast to this study that documented precursors in the SVZ, a study found numerous clearly BrdU-labeled cells in the brains of adult squirrel monkeys (*Saimiri sciureus*) that were raised in an enriched environment for three weeks and were untreated; these were found mainly in the dorsal and ventral striatum, including the nucleus accumbens, and were less abundant in the caudate nucleus and putamen. Additionally, double staining for markers BrdU/NeuN revealed the presence of mature cells in the striatum [143]. Similarly, a study found that new neurons in the striatum of adult rabbits are generated from precursor cells located in the caudate nucleus, where these cells express early neuronal markers, such as DCX, polysialylated neuronal cell adhesion molecule (PSA-NCAM), β-tubulin class III (TuJ) and HuC/D protein, and later, these neuroblasts migrate and differentiate into striatal spiny interneurons. Thus, neurogenesis in the striatum of adult monkeys and rabbits is independent of adjacent SVZ neurogenesis [144].

Adult striatal neurogenesis in the human brain has been confirmed using a technique that retrospectively determines the date of cellular birth in humans. This technique, developed by Ernest and his group of collaborators (2014), is based on detecting changes in the levels of the carbon-14 isotope (^1^⁴C) in the DNA of proliferating cells. They used accelerator mass spectrometry, which revealed that the levels of the ^1^⁴C in genomic DNA closely parallel atmospheric levels that were generated during the cold war atomic age, to determine the time point when DNA is synthesized and cells are born in areas of the adult human brain, and showed postnatal cell turnover in the striatum [145]. Additionally, the transcriptome data from many adult human brains determined that DCX was mainly expressed in the adult human striatum rather than in the hippocampus [146]. This finding was corroborated with other techniques, such as Western blotting and immunohistochemistry, with other different neuroblast markers, such as PSA-NCAM in the postmortem human brain, where the authors found the same number of neuroblasts in the striatum, SVZ and hippocampus in the human brain. Furthermore, Ernest and colleagues identified that the cells of the human striatum are devoid of lipofuscin, an age pigment, suggesting that they represent young neurons. Additionally, the authors examined human patients with Huntington’s disease (HD) and observed that postnatally generated neurons in the striatum were depleted in patients with the advanced stages of the disease [145].

Moreover, a recent study found that stem cell-derived human striatal progenitors grafted into a rat model of HD mature in vivo, and several differentiated into medium spiny neurons that integrated into local circuits of transplanted and host cells. These neurons formed extensive fibers that projected toward appropriate striatal targets where cell differentiation and integration permitted the alleviation of sensorimotor deficits generated by HD in a rat model, although experiments at longer time points are needed to confirm the ability of the graft and its function [147]. Similarly, ectopic coexpression of microRNAs such as miR9/9* and miR-124, together with the transcription factors BCL118, DLX1, DLX2 and MYT1L, promoted direct conversion of human postnatal and adult fibroblasts into population medium spiny neurons in the striatum. Additionally, a study showed that when transplanted in the mouse brain, reprogrammed human cells persisted over six months, exhibited characteristics equivalent to native neurons, and extended projections to anatomical targets such as the striatum [148].

Therefore, the function of adult human striatal neurogenesis remains to be established. The longevity of adult-born neurons argues for probable functional integration that can be used for therapeutic purposes in patients with striatal disorders such as Huntington’s disease, Parkinson’s disease (PD), Alzheimer’s disease (AD) and other disorders [132].

Overall, in adult striatal neurogenesis, some neuronal stem/progenitor cells (NSPCs) become restricted to the SVZ and migrate to the impaired striatum of nonhumans and humans for differentiation into interneurons. However, few studies have examined the molecular mechanisms underlying neurogenesis induced by pathological conditions that potentially regulate proliferation, migration, and differentiation in the striatum. Li and colleagues (2021) recently described similar epigenetic mechanisms that participate in adult neurogenesis post-Alzheimer’s disease. They explored the effects of AD7c-NTP silencing following AD injury. AD7c-NTP was associated with AD neurodegeneration, and the silencing was mediated, in part, by MeCP2 phosphorylation at serine 421 (S421) coupled to DNA demethylation in the *Gfap*, *Nestin* and DCX promoters, preventing MeCP2 from binding to its cellular target, and thereby decreasing transcriptional repression to induce gene expression. These gene promoters may be implicated in the regeneration and fate determination of NSPCs in adult striatal neurogenesis [149].

However, other molecular mechanisms are poorly understood but are important because they regulate the transient increase in NSPC proliferation in the SVZ and the migration of neuroblasts toward the damaged area after stroke, and are endogenous negative regulators, such as the LNK protein, which is expressed in NSPCs in the SVZ of adult rodents and humans. Here, when the LNK protein is expressed at low levels, NSPC proliferation is increased in the SVZ through STAT1/3 transcription factors. However, when the LNK protein is overexpressed, it attenuates the insulin-like growth factor (IGF-1) signaling pathway through the inhibition of AKT phosphorylation, which induces a reduction in the proliferation of NSPCs [150]. Another endogenous negative regulator of stroke-induced proliferation of NSPCs in the SVZ is tumor necrosis factor receptor-1 (TNFR-1). The increase in cell proliferation over one week after stroke was related to the increases in the number of microglia and the expression of the TNFR-1 and TNF-α genes in the SVZ. The blockade of TNF-R1 signaling might promote the proliferation of cells in the SVZ, and neuroblast formation is enhanced [151].

Regarding the migration process in the adult striatum of animals and humans, different studies have identified several factors that participate in migration after stroke. Some of these factors include stromal cell-derived factor 1 (SDF-1) and its receptor CXCR-4 [152,153], monocyte chemoattractant protein 1 (MCP-1) and its receptor CCR2 [152,153], and the extracellular proteases matrix metalloproteinase 9 (MMP-9) and MMP-1 [152,153,154], among others. A study reported shorter survival and differentiation of new neurons in the striatum of adult rodents after stroke (151). However, in the injured striatum of adult humans, differentiation and survival occur at low levels over long periods [136]. Recent studies have attempted to understand the mechanism that controls this process, as well as the role of these new cells in adult neuronal functioning and the potential therapeutic benefits that these cells may provide (see Table 5 for a summary of the material presented here).

### 3.4. Adult Neurogenesis in the Amygdala: Mechanism and Possible Functional Implications

The amygdala is one of the regions that forms the limbic system. The amygdala is responsible for processing vital emotions, such as fear learning and memory, among others. According to recent reports, this structure generates new neurons throughout life. An experiment carried out with nine adult squirrel monkeys and four adult cynomolgus monkeys, which were injected i.v. with BrdU twice a day for 3 days and euthanized at different time points, provides evidence supporting this claim. Another group of squirrel monkeys received a left lateral ventricular injection with the dye 1,1′-dioctadecyl-3,3,3′,3′-tetramethylindocarbocyanine (Dil) and was euthanized at the third week postinjection. The research group reported newly differentiated cells in the amygdala at 21 and 28 days postinjection [158]. The amygdala and hippocampus play important related roles in the limbic system. However, in terms of neurogenesis, they function differently. A study conducted using 8-week-old male Wistar rats in which the olfactory bulb was removed showed decreased adult neurogenesis in the hippocampus but increased neurogenesis in the amygdala 3 weeks after bulbectomy. This increase was independent of treatment with imipramine, as reported in a parallel cohort of animals [159]. Additionally, adult neurogenesis in the amygdala in response to hormones has been reported. Fowler et al. (2003) evaluated the effect of testosterone and its metabolites on newly generated cells in the amygdala of adult male meadow voles (*Microtus pennsylvanicus*) [160]. The authors reported that castrated males treated with testosterone propionate or with estradiol benzoate exhibited an increased number of BrdU-positive cells in the amygdala, but not in the hippocampus or the hypothalamus. The time course data showed that the hormonal effect begins at 30 min [160]. Newly generated amygdalar cells have a neuronal phenotype because they express Tuj1, an early neuron marker. Interestingly, socialization plays an important role in the generation of neurons in the amygdala during adulthood in rodents. Female prairie voles that had the opportunity to socialize and mate for 48 h with males had more BrdU-positive cells than isolated females [161]. These changes persisted for over 3 weeks and were region-specific because the increase in the newly generated pool was only observed in the amygdala but not in the caudate/putamen or the hippocampus [161]. The effect of socialization on amygdala neurogenesis was also observed in other rodents. Female C57BL6/J mice living under environmental enrichment for 40 days showed an increase in the expression of proteoglycan neuron-glia 2 (NG2), a marker that represents parenchymal precursor cells [162]. This increase was also induced by voluntary running [162]. Due to the relevance of the amygdala in emotional regulation, the participation of stress in neurogenesis in this region must be characterized. Saul et al. proposed a decrease in the pool of NG2/BrdU cells under stress [163].

Amygdalar neurogenesis is a topic with several open questions that would be very interesting to answer, such as the effect of social stress or a potential relationship between the ventral hippocampus and neurogenesis in the amygdala due to its functional relationship with emotions. More focused research is needed to assess this critical area of the limbic system.

## 4. The Controversial New Adult Neurogenesis Zones

Several of the new neurogenic sites have already been reported to have processes that are considered incomplete in terms of their final result. In addition, the diversity was determined using various measures, including regional location, progenitor identity, and progeny fate. In the study by Feliciano et al., these aspects strictly depend on the animal species, suggesting that persistent neurogenic processes have uniquely adapted to the brain anatomy of different mammals [164]. Tissue growth may be a critical feature in the regulation of adult neurogenesis [165]. As an example, teleost fish species with indeterminate growth also exhibit neurogenesis throughout life, as well as amazing brain repair and regeneration capabilities [166]. Ponti et al. suggested that undetected adult gliogenesis and neurogenesis might exist in mammalian species with long lifespans and slow growth, including primates and humans [165].

Another point that emerges from the studies described above that is important for the study of new neurogenic sites and to obtain conclusive evidence is the technical question. Some discrepancies or differences between the different studies by several authors, such as Feliciano et al. [164] or Kempermann et al. [167], are mainly attributed to technical problems. Various alternatives to performing a proliferation and neurogenesis study must be considered from the selection, dosage, and administration protocol of proliferation markers (such as BrdU and IdU), the selection of neuronal markers for colocalization, fixation type and duration, and problems with the detection of marker proteins associated with the fixation, the primary and secondary antibodies to be used, and the microscopy equipment, among many other factors. Due to the great diversity of technical problems, much of the variability in the reports from the different research groups may likely be attributed to the lack of optimization of the protocols used for the detection of neurogenesis, rather than to the absence of neurogenesis itself.

We present two of the most criticized neurogenic zones in mammalian brains, the habenula and the cerebellum (see Table 6, which provides a summary of the material detailed in the text). We described the existing evidence of adult neurogenesis in mammals, from constitutive or spontaneous neurogenesis to processes that occur after pharmacological or genetic manipulation.

### 4.1. Adult Neurogenesis in the Habenula: Mechanism and Possible Functional Implications

The habenula is a bilateral brain structure that is part of the dorsal diencephalic conduction system [175], which is composed of the stria medullaris (SM), fasciculus retroflexus (FR), and habenular nucleus (Hb) [176]. Its name is derived from the Latin word habena, which means “little rein”, based on its particular morphology. The Hb links the forebrain and several brainstem regions. The Hb is present in all vertebrates, revealing its archaic origin. The mammalian habenula is often divided into the lateral habenula (LHb), medial habenula (MHb) [177], and habenular commissure [176]. The LHb receives inputs from the basal ganglia, hypothalamus, and limbic regions that integrate information from within the organism and the current external environment. This information is subsequently sent to several brainstem regions with ascending projections that assist in modulating and updating behavior to adapt to an ever-changing environment. Thus, the LHb plays a key role in learning to inhibit distinct responses to specific stimuli (for more comprehensive information, see Sosa et al., 2021 [178]).

The presence of neurogenic niches in the habenula has been reported in different animals, from teleost fish—which are considered to have adult stem cells [28,179,180,181,182,183,184]—to rodents [29,157]. For example, in vertebrates such as zebrafish, proliferation zones in the fish telencephalon, habenula, and hypothalamus were reported via the observation of BrdU-labeled cells [181]. Secondary evidence that the habenula is a neurogenic niche has been elucidated in studies related to diverse transcription factors associated with the modulation and control of adult neurogenesis. One of these proteins is the protein prothymosin alpha (ProTα), an acidic nuclear protein implicated in several cellular functions, including cell cycle progression, proliferation, and survival. ProTα is expressed in brain regions that are relevant to neurogenesis, including the SVZ, the granular cell layer of the DG, and the granule cell layer of the OB. Interestingly, strong immunoreactivity was also detected in the habenula, which, as we have discussed, is a neurogenic niche. The author speculated that ProTα expression is related to neurogenesis [185]. In more recent research, AHN was significantly decreased in heterozygous ProTα KO mice [67]. Additionally, the authors observed the downregulation of several neurogenesis-related genes, such as Nrp1 and Nrxn3, in heterozygous ProTα KO mice. The authors suggested that ProTα regulates the expression of candidate genes involved in adult neurogenesis [67]. This finding might be relevant in brain regions ranging from the hippocampus to other neurogenic zones. Another transcription factor linked to neurogenesis is NeuroD1, also known as NeuroD or β2. It is a member of the proneural gene family that plays an important role in embryonic neurogenesis as a neuronal differentiation factor; e.g., it is upregulated during postmetamorphic neurogenesis [186]. NeuroD1 gene activity was detected in various brain regions of adult *Xenopus laevis*, an African aquatic frog with a high proliferation rate and a high rate of new cell production in the adult brain. The most densely NeuroD1-labeled cell cluster was located in the epithalamus and the dorsal and ventral nuclei of the habenula [186].

Regarding studies conducted in mammals, Sachs and Caron found that the administration of chronic fluoxetine (in drinking water at 155 mg/L for 4 weeks), a selective serotonin reuptake inhibitor, to adult male mice significantly increased BrdU incorporation in the medial habenula [29]. The authors suggested that fluoxetine enhances neurogenesis in the habenula by increasing proliferation rather than altering the survival of newly generated cells or the percentage of dividing cells that commit to a neuronal lineage. Additionally, fluoxetine leads to increased gliogenesis in the habenula. Brain-derived neurotrophic factor (BDNF) mRNA levels were significantly increased in the habenula, indicating that BDNF might be involved in the effects of fluoxetine on cell proliferation and neurogenesis in this region [29]. The authors suggested that the increase in BDNF expression plays a role in the increase in cell proliferation within the medial habenula, as previously described by Pencea et al., where BDNF promoted cell proliferation and neurogenesis in the hypothalamus and the habenula [157]. Since the habenula mediates responses to stressful and aversive stimuli [187,188], Sachs and Caron suggested that neurogenesis in this structure potentially plays an important role in buffering stress responses and in mediating behavioral responses to the antidepressant fluoxetine [29].

In an experiment by Pencea et al. with adult Sprague-Dawley rats, a BDNF infusion in the lateral ventricle resulted in numerous BrdU-positive cells in the Hb (in addition to the SVZ, striatum, septum, thalamus, and hypothalamus) [157]. Interestingly, no BrdU-positive cells were observed in the part of the DG immediately adjacent to the third ventricle. Furthermore, tropomyosin receptor kinase B (TrkB, a receptor for BDNF) is expressed at a uniformly high level throughout the Hb. Nevertheless, TrkB expression is not sufficient for cell proliferation, since BrdU-positive cells are much more numerous along the medial edge of the habenula. Akle et al. reported a similar result in the habenular neurogenic niche in zebrafish, where the majority of BrdU-positive cells were located along the midline of the habenula, adjacent to the diencephalic ventricle and ventral nucleus of the habenula [179]. Pencea et al. argued that this disparity could not be accounted for by the differences in BDNF exposure because the dorsal edge of the Hb, which showed lower BrdU incorporation, also faces the third ventricle [157]. Finally, the authors concluded that TrkB expression correlates with the level of BrdU expression [157]. This relation potentially suggests that the TrkB receptor for BDNF mediates cell proliferation in the habenula and other structures.

However, evidence of spontaneous or constitutive adult neurogenesis in the mammalian habenula is not available. A quite plausible hypothesis is that this phenomenon occurs naturally in response to demanding internal and external changes. Adult neurogenesis occurs after pharmacological treatments (antidepressants) and neurotrophin infusion (BDNF). The habenula is present in all vertebrates, revealing its archaic origin. In the vast majority of these species, adult neurogenesis has been confirmed. The LHb plays a key role in learning to inhibit distinct responses to specific stimuli [178]. All the above findings suggest that constitutive adult neurogenesis in the habenula could be an adaptive mechanism. In addition, it occurs in other structures with strong responses to the environment that participate in different learning processes, such as the hippocampus, olfactory bulb, and striatum. A plausible speculation is that technical problems are another potential interfering factor. More research is needed to answer the question of whether adult neurogenesis occurs in the mammalian habenula under normal physiological conditions and to reveal the underlying mechanisms of the adult neurogenic process.

### 4.2. Adult Neurogenesis in the Cerebellum: Mechanism and Possible Functional Implications

As described above, neurogenesis occurs in several brain regions, mainly in the SGZ of the hippocampus and SVZ of the lateral ventricle [31,189]. However, evidence of this brain mechanism has been reported in other structures, such as the cerebellum, a complex region involved in several processes in addition to coordination and motor control, including perception, emotion, and cognition [190,191,192]. This structure originates from rostral metencephalic vesicles and caudal mesencephalic vesicles, specifically from the alar plate of rhombomere 1, and it is located at the anterior end of the hindbrain [193,194]. The cerebellum is tightly organized as a trilaminar structure consisting of an outer molecular cell layer, a middle Purkinje cell layer, and an inner granular cell layer [195]. Through the different layers, several types of cells constitute the cerebellum, including granular cells, Purkinje cells, unipolar brush cells, deep cerebellar neurons, various interneurons, and glial cells [196]. The granular layer consists of different types of interneurons: granule cells and unipolar brush cells, which are excitatory [197], and Golgi and Lugaro cells, which are inhibitory [198]. In the molecular layer, basket and stellate cells and inhibitory interneurons are besieged [199]. Finally, in the Purkinje layer, homonymous GABAergic cells constitute the sole output of the cerebellar cortex [197]. The development of the cerebellum starts at embryonic stages and continues postnatally; in humans, the development of this structure extends from 35 to 42 embryonic days until the second year [200]. Projection neurons are generated first; then, at the late embryonic and early postnatal stages, local interneurons are born [197,201]. Several molecules are involved throughout cerebellar development and are mainly driven by the isthmic organizer.

Cerebellar neurogenesis is a complex process that includes the strategic, temporal, and spatial interaction of multiple molecules, constituting the different phases of birth, migration, and maturation of the cells that compose the complex cerebellar functional network. In mammals, this cerebellar neurogenic process occurs primarily during embryonic development and continues after birth for a short period [164,202]. However, in transgenic mice, evidence of adult neurogenesis has been obtained even after cerebellar injury; mainly, restitution of the granular outer layer is possible after complete or partial damage, again involving the previously described processes [201,203]. In addition to genetic manipulation, other conditions have been established to promote cerebellar adult neurogenesis. Transplanting human cerebellar granule neuron precursors into the Harlequin mouse cerebellum successfully triggers the proliferation of endogenous nestin-positive precursors (even when transplanted cells did not survive), which differentiated into mature cells [204]. NSCs were identified beyond lateral ventricle walls and the dentate gyrus and, interestingly, in the cerebellar tissue of mice [171]. NSCs can be derived from both neurons and glial cells, a differentiation process that is regulated by transcription factor family Sox genes, in particular Sox1, Sox2, and Sox9 [172,205]. The expression of the transcription factor Sox2 in the mouse cerebellum drives Bergmann Glia development during embryonic, postnatal, and adult neurogenesis processes [171,196]. Sottile et al. (2006) observed that Sox1- and Sox2-positive Bergmann glia formed a distinct intercalated pattern with calbindin-positive Purkinje cells. The authors concluded that their results showing that three Sox genes that mark NSCs are expressed in adult Bergmann glia suggest that Bergmann glia represent an NSC-like population that was as-yet uncharacterized in the adult cerebellum and identified in their study by the expression of Sox1, Sox2 and Sox9 [171]. Nevertheless, Sottile et al. recommend an in vitro evaluation of the intrinsic neurogenic potential of Bergmann glial cells in the absence of local factors that might bias differentiation toward glial phenotypes. They hypothesize that if the Sox1/Sox2/Sox9-positive Bergmann glial population lacks stem cell features, their evidence would imply that these transcription factors are not broad stem cell markers in the adult brain. Instead, they might be more likely to be associated with the radial glia phenotype itself, which is also shared by other NSC populations [171].

Adult cerebellar neurogenesis was reported in transgenic mice exposed to physical activity and an EE, as detected by the expression of Sox2 in the Purkinje cell layer [206]. Bergmann glia have an important role in modulating Purkinje cell signaling since they are responsible for glutamate uptake and potassium homeostasis [207]. This function might be one reason why this neurogenic glial population develops in response to motor activity demand.

In addition to these experimental manipulations and the use of transgenic mice, adult neurogenesis has been detected in the cerebellum of New Zealand White peripubertal and adult rabbits [165,208]. Neurogenesis occurs even though the outer granular layer disappears in adulthood and is replaced by a proliferative layer called the “subpial layer”, which persists beyond puberty on the cerebellar surface [165]. The Purkinje cell layer incorporated BrdU at 1–5 days postinjection survival, and incorporation of BrdU was also detectable in some Bergmann glial cells. Double staining for BrdU with PSA-NCAM and Map5 revealed that a large population of cells expressing these markers in the peripubertal rabbit cerebellar cortex were newly generated. Additionally, the authors found DCX+ or Map5+ and BrdU+ cells at 15 days of survival after the injection, revealing that both cell populations in the adult cerebellar cortex were newly generated [165]. They also reported the neurogenesis of GABA+ cells immunoreactive for Pax2, a marker for GABAergic cerebellar interneurons of neuroepithelial origin. The authors concluded that a subset of GABAergic interneurons are generated within the molecular layer of the peripubertal rabbit cerebellum [165].

Based on the evidence described above, neurogenic events occur in the cerebellum beyond the early stages of mammalian life. More research is needed to reveal the processes and environmental signals associated with regulating the neurogenic process. Importantly, environmental mediation and cognitive demand are fundamental factors that promote the neurogenic process in adults in brain structures, such as the hippocampus, OBs, and prefrontal cortex. Nevertheless, at the cerebellar level, the neurogenic process remains uncertain.

## 5. MicroRNA Modulation of Human Adult Neurogenesis

MicroRNAs (miRNAs) are small (≈22 nucleotides), single-stranded noncoding RNA molecules that regulate gene expression posttranscriptionally through complementary binding to untranslated regions of target messenger RNA (mRNA) targets. Thus, miRNAs inhibit protein synthesis [209,210,211]. Recently, miRNAs have been described as crucial regulators of the modulation of embryos through adult neurogenesis [209,210,211].

Primary miRNAs are recognized and processed by the RNase III enzyme Dicer, which cleaves premiRNAs in the cytoplasm to generate functional, mature miRNAs. Then, miRNAs suppress specific mRNAs by guiding the RNA-induced silencing complex (RISC) to complementary target sites and beginning the RNA polymerase II-mediated transcription of long primary miRNAs in the nucleus [211]. According to previous studies, miRNAs are implicated in tissue morphogenesis and some cellular processes, such as apoptosis, developmental timing, differentiation, and myogenesis [212]. As mentioned above, they are also expressed at high levels in the central nervous system (CNS) and play an essential role in stem cell proliferation and differentiation [213]. NSCs lacking the RNase III enzyme Dicer were incapable of differentiating into both glial and neuronal fates, although they were able to proliferate [214]. Therefore, miRNAs are clearly epigenetic regulators that must be studied in depth to more clearly understand the implications of the various cellular phenomena that are also altered by internal or environmental changes. Recently, the expression of miRNAs was reported to be modified by internal or external changes, such as stress, glucocorticoids, and pharmaceutical drugs, functioning as mood stabilizers [215]. Adult neurogenesis can be modulated by various factors, such as social-environmental events that affect neuronal function and behavior.

Below, we present evidence of the modulatory effect of miRNAs on adult neurogenesis in different neurogenic niches in the brain. Since the aim of this article is not to provide an in-depth description of, but only to exemplify, the importance of the modulatory role of miRNAs on adult neurogenesis, we suggest that the reader review the recommended articles below that address the topic in-depth (see [91,210,216]).

Studies have reported that miRNAs have a role in mediating adult neurogenesis. For example, the miR17-92 cluster is substantially upregulated in neural progenitor cells of adult mice after stroke; in particular, the overexpression of the miR17-92 cluster in the SVZ of ischemic animals increased cell proliferation and survival [156]. Additionally, a study by Liu et al. revealed that stroke altered the expression of multiple miRNAs, substantially reducing the expression of miRNA-124a, a neuron-specific miRNA, in neural progenitor cells of the SVZ; in this study, the introduction of miR-124a inhibited ischemia-induced neural progenitor cell proliferation and promoted neuronal differentiation of progenitor cells [155]. Notably, miR-124 was found to dictate postnatal neurogenesis in the mouse SVZ; specifically, it was shown that miR-124 activity is rapidly initiated once NSCs, type B cells, transit to rapid-amplifying progenitors, type C cells [217]. Additionally, in the SVZ, signaling molecules, such as bone morphogenic proteins (BMPs) and their secreted inhibitor Noggin (a protein that is involved in the development of many body tissues), play important roles in controlling the behavior of NSCs. In this regard, Noggin expression significantly increases neuronal and oligodendrocyte differentiation in vivo and in vitro (using neurospheres), suggesting that Noggin/BMP interactions tightly control cell fate in the SVZ [218]. Another miRNA, miR-410, is downregulated during the inhibition of BMP signaling by Noggin. Although overexpression of miR-410 in SVZ neurospheres inhibited neuronal differentiation and increased the number of astrocytes produced, the loss of miR-410 function exerted the opposite effect, promoting neuronal differentiation at the expense of astrocyte formation; furthermore, miR-410 has been suggested to function downstream of BMP signaling when coexpressed with Noggin, reducing the increase in Noggin-induced neuronal differentiation to control the levels. Based on these findings, miR-410 may provide a new mechanism for the essential choice of NSCs between self-renewal and differentiation [219].

According to previous studies, miRNAs are essential for normal brain development and for establishing the functional connectivity of the brain and regulating adult hippocampal neurogenesis [220]. In this regard, miR-132 was recently shown to promote neurogenesis in both the embryonic nervous system and adult brain [221]. In particular, Walgrave et al. showed that miR-132 regulates neurogenesis in adult mice and the human brain and that it is also necessary for the formation of excitatory synapses [222]. Additionally, miR-132 deletion impaired memory and modified the hippocampal transcriptome [223], while the overexpression of miR-132 in cultured neural stem cells of the adult rat DG enhanced their differentiation [221]. These results suggest that miR-132 is a locus that regulates cognitive capacity and neurogenesis in the hippocampus. Another microRNA known to participate in neurogenesis is miR-124, which is abundantly expressed in neurons, involved in neural differentiation and necessary for AHN [224]. Furthermore, the regulation of miR-124 in AHN may also be associated with Notch, REST/SCP1, and other signaling pathways [225]. Additionally, Mojtahedi et al. showed that voluntary treadmill running increased miR-124 and miR-132 expression while reducing the expression of their respective targets, glucocorticoid receptor, SOX9, and protein P250 in adult male rats. Meanwhile, an increase in cAMP-response element-binding protein (CREB) was observed. Accumulating evidence suggests that voluntary running increases adult neurogenesis [226].

Regarding other structures, epigenetics refers to the effects of the environment on phenotypic expression; specifically, in the cerebellum, DNA methylation mediated by the expression of 5-lipoxygenase is associated with the modulation of the proliferation and differentiation of cerebellar precursor cells [174]. Similarly, the enriched miR-592 plays an important role in the differentiation of neuronal stem cells, regulating the morphology of neurons and astrocytes in the rat cerebellum, while its silencing disturbs neuronal maturation [227]. Drug abuse also contributes to epigenetic changes, and early exposure to ethanol interferes with microRNA expression, promoting apoptosis specifically by suppressing miR-29b expression [228].

Additionally, several studies have documented the important role of miRNAs in neurogenesis induced by pathological conditions. These miRNAs are small noncoding RNA molecules of 22 nucleotides that regulate gene expression [229]. Interestingly, miR-124 has a neuroprotective function in HD because it slows the progression of HD in transgenic mice, increases cell proliferation in the SVZ and promotes neuronal differentiation in the striatum of the mouse brain. miR-124 also upregulated PGC-1α and BNDF expression and downregulated SOX-9 expression in the striatum of HD mouse brains, which implies increased neuronal survival and differentiation in the striatum [230]. Likewise, Liu and collaborators (2011) found that experimental stroke altered the expression of miRNAs in adult rodents, including a reduction in the expression of miR-124a that activates the Jagged1/Notch1 signaling pathway, thus promoting an increase in the proliferation of NSPC in the SVZ and promoting neuronal differentiation [156,231,232]. Another study showed that members of the miR-17-92 cluster were upregulated after ischemia. Overexpression of the miR-17-92 cluster increases the proliferation and survival of NSPCs in the SVZ by reducing the expression of the target phosphatase and tensin homolog (PTEN), while the Sonic hedgehog (Shh) signaling pathway recruits N-MYC to regulate miR-17-92 cluster expression in NSPCs [156].

The regulatory role of miRNAs is very important in adult neurogenesis, as well as in many neurodegenerative disease states. Thus, more research is needed to understand the patterns of miRNA expression and their correlations with neurogenic stages, disorders, and diseases. As we described, miRNAs modulate multiple targets and their functions. Thus, miRNAs have the potential to serve as useful diagnostic biomarkers for those disorders and diseases, as well as for developing novel therapeutic strategies.

## 6. Neurotrophic Factor and Neurotrophin Modulation of Adult Human Neurogenesis

Neurotrophic factors (NTFs) are proteins that regulate the survival, growth, morphological plasticity and synthesis of proteins required for differentiated functions of neurons [233]. NTFs are grouped into three families: neurotrophins, glial cell line-derived neurotrophic factor (GDNF) family ligands (GDNF-family ligands or GFLs) and neuropoietic cytokines. The pathways triggered by each of these families are functionally distinct; however, cellular responses overlap [234]. The most studied and the first NTF family identified was neurotrophins (NTs). NTs are a family of pleiotropic molecules implicated in performing important roles in the regulation of neuronal differentiation and survival during development, axonal and dendritic growth, synaptic transmission, and adult neural plasticity in adults, with key roles in memory and learning processes [235]. The first described NT was nerve growth factor (NGF) [236], and other family members were subsequently identified, such as BDNF, NT3, and NT4/5 [237]. First, NTs are synthesized as precursors or proneurotrophins (pro-NTs), which may be secreted from cells or continue to undergo intracellular proteolytic cleavage to yield mature neurotrophins [235]. Two classes of receptors for NTs have been identified, the common p75 neurotrophin receptor (p75NTR) and the Trk receptor tyrosine kinase family. To date, three different Trks have been identified in mammals: TrkA, TrkB, and TrkC. In general, TrkA preferentially binds the ligand NGF, TrkB selectively binds BDNF and NT4/5, and TrkC preferentially binds NT3. All NTs bind to p75NTR, and while only mature neurotrophins bind to Trk receptors, pro-neurotrophins also interact with p75NTR [237] (Vilar, 2016). NTs and their receptors are implicated in the regulation of adult neurogenesis [237].

The GDNF family of ligands (GFLs) are synthesized, secreted and activated by diverse tissues. GFLs bind to receptors on target cells, modulating development, survival and differentiation. GFLs are produced in the form of a precursor (preproGFL) and then proteolytically cleaved to proGFL [238]. The GFL family members identified to date are GDNF, neurturin (NRTN), artemin (ARTN), and persephin (PSPN). GFLs signal through receptor tyrosine kinase (RET), but are activated only if the GFL is first bound to GDNF family receptor-α (GFRα) receptors. Four different GFRα receptors have been characterized to date (GFRα1–4), which determine the ligand specificity of the GFRα–RET complex. GFRα1 preferentially binds to the ligand GDNF and then forms a complex with RET. GFRα2 selectively binds to NRTN, ARTN activates RET by binding to GFRα3, and PSPN binds to GFRα4 [238,239]. GFLs play interesting roles in different tissues, including survival, differentiation and migration. In particular, signaling by GDNF promotes the survival of dopaminergic neurons; in fact, GDNF is absolutely required for the survival of dopaminergic and noradrenergic neurons in the adult brain [240].

Finally, neuropoietic cytokines (NCs) are small proteins first characterized as components of the immune response, but we now know they play a much broader role in various physiological features [241]. This family consists of interleukin-6 (IL-6), IL-11, IL-27, leukemia inhibitory factor (LIF), ciliary neurotrophic factor (CNTF), cardiotrophin 1 (CT-1), neuropoietin, and cardiotrophin-like cytokine (CLC), also known as novel neurotrophin 1 (NNT1), and B cell stimulating factor 3 (BSF3) [241]. Below, we present evidence of the modulatory effect of NFTs and NTs on adult neurogenesis in different neurogenic niches in the brain. They have been implicated in the regulation of adult neurogenesis in several brain structures, including the SVZ and SGZ of the DG, SN, striatum, habenula, and cerebellum.

BDNF is a small protein (252 amino acids) that is abundantly expressed in the nervous system. Evidence of the participation of BDNF and its receptor TrkB in adult neurogenesis is abundant and the subject of several studies. Some reports indicate that BDNF also plays a role in neurogenesis within the SVZ, although its precise role remains controversial. Most studies show that BDNF does not promote significant changes in cell proliferation and survival, but it has an important role in the migration of SVZ-derived cells. Additionally, through TrkB (the BDNF receptor) signaling, BDNF was shown to have an essential role in regulating the dendritic complexity and synaptic formation, maturation, and plasticity of newborn neurons [34]. In this regard, a study by Chiaramello et al. shows that BDNF and its receptor TrkB are expressed throughout the migratory pathway in vivo, implying that BDNF might mediate migratory signaling [35]. Furthermore, studies have indicated that BDNF may regulate neurogenesis after a neurological event, such as subarachnoid hemorrhage (SAH) or ischemia. In a study by Lee et al., BDNF was identified as a key factor associated with neurogenesis after SAH [36]. Additionally, researchers detected increased BDNF concentrations in cerebrospinal fluid on days 5 and 7 after inducing SAH in a rat model. Additionally, the authors reported that BDNF expression was upregulated in the SVZ of rats on days 5 and 7 post-SAH and that NSCs, astrocytes, and microglia in the SVZ released BDNF. Taken together, BDNF is an important regulator of neurogenesis within the SVZ after SAH [36]. Another study by Luo et al. (2017) found that 20 Hz repetitive transcranial magnetic stimulation (rTMS) activated the BDNF/TrkB pathway in rats after MCAO; activation of this pathway led to beneficial effects that significantly promoted neurogenesis within the SVZ [242].

NTs are some of the regulators of neurogenesis that occurs in response to exercise and environmental enrichment, with BDNF being particularly important [243]. Additionally, a key factor in the neuronal response to enrichment is the release of BDNF and the activation of the mitogen-activated protein kinase (MAPK) cascade, which can lead to the stimulation of neurogenesis [64]. Isolated in vitro, new hippocampal cells expressing Prominin-1 are unique and fulfill every criterion to be considered stem cells: proliferation, self-renewal, and multipotentiality [244]. Thus, BDNF mediates the effects of these cells by activating several intracellular pathways, such as MAPK [65], leading to an increase in cAMP-response element-binding protein (CREB) expression [66] and promoting B-cell lymphoma-2 (Bcl-2) synthesis [245]. Additionally, an increase in the BDNF/CREB/Bcl-2 regulatory pathways underlies a molecular basis for AHN [243], synaptic plasticity and memory processes [246]. Furthermore, postnatal BDNF-TrkB activation in immature DG cells enhances their sequential maturation; thus, BDNF modulates AHN [63]. GDNF is another neurotrophic factor proposed to be involved in hippocampal neurogenesis [247]. GDNF is the prototypic member of a small family of neurotrophic factors that promote cell survival, outgrowth, and differentiation of distinct populations of neurons (central and peripheral) along with their development [248]. Recently, Bonafina et al. showed that GDNF is critical for the morphological maturation and synaptic integration of adult-born neurons in the DG and is necessary for the memory process [249]. Additionally, GDNF facilitates the differentiation of neurons in the DG through a signal transducer and activator of transcription 3 [250]. On the other hand, Zhang et al. showed that conditional knockout of GDNF led to reduced adult neurogenesis in the hippocampus [251], while an infusion of GDNF in this region in rats increased neurogenesis within the DG [252].

Regarding the SN and NTFs, much more research is needed on this matter. One day after a lesion was created in the SN through local MPP administration, an injection of BDNF (100 ng) diminished the number of BrdU-positive cells in the SN [122]. Based on this result, the relationship between NTFs and neurotoxins is more complex than initially presumed. NTFs have been described in the SN, especially those related to DA. One of the main NTFs is Shh, a member of the Hedgehog family (for a further review, see [253]). Shh produced by dopaminergic neurons in the SNpc maintains homeostasis through a noncell autonomous process involved in cellular differentiation, maintenance and survival [128]. Ortega-de San Luis et al. performed a series of experiments to suggest that NTFs are present in the SN. The number of dopaminergic neurons in the SN (diminishing the expression of Shh and other NTFs) was depleted through the deactivation of the mitochondrial Sdhd gene in mice [129]. Animals showed a regular number of DA neurons at birth but suffered DA neurodegeneration postnatally. By comparing this animal model to animals with normal DA neurons, the authors found a decreased number of striatal fire-spiking (FS) GABAergic interneurons in the experimental group. This result shows a clear relationship among DA, the NF Shh, and other systems, such as the GABAergic system.

After intraventricular infusion of BDNF in adult rats, new neurons were generated not only in the SVZ but also in other brain areas, such as the parenchyma of the striatum [157]. Additionally, several NTs have been implicated in controlling survival and differentiation during adult postnatal SVZ neurogenesis. BDNF promotes the survival and differentiation of NSPCs and neuroblasts in the rat striatum after stroke or injury. In addition, the intraventricular infusion of BDNF in rats generated an increase in the numbers of newly born neurons and affected migration to the striatum [130]. In addition, adenoviral overexpression of BDNF following QA lesioning of the striatum recruits SVZ-derived cells to the site of injury [254]. In another study, an infusion of BDNF in combination with vascular endothelial growth factor (VEGF) increased SVZ neurogenesis in mice after stroke [252]. However, the participation of other NT family members in adult striatal neurogenesis has also been documented. For example, the intraventricular administration of NGF, which increased the number of proliferating cells in vivo in the SVZ of aged mice [255], and intranasal administration of NGF to adult rats after cerebral ischemia may increase the survival of newly formed neurons and promote striatal neurogenesis. These new neurons may integrate into lesioned circuits and improve neurological functions [170]. Furthermore, growth factors have also been implicated in adult neurogenesis; for example, a study found that VEGF plays an important role in regulating the proliferation of cells in the SVZ and the migration of neuroblasts in the striatum after stroke [152,153].

As we reported above in the section discussing the habenula, significantly increased BDNF mRNA levels have been reported in the habenula after fluoxetine infusion. The authors suggested that the increase in BDNF expression plays a role in increasing cell proliferation within the medial habenula [29]. NTs play important roles at different stages of cerebellar cellular differentiation. NGF is related to the survival of Purkinje cells, while BDNF and NT-3 are present in the immature cerebellum [173]. However, after birth, the expression of these molecules changes; NT-3 reaches its maximal level just after birth, while NGF expression peaks 2 weeks thereafter [256]. This difference is because NT-3 does not promote cell survival. Nonomura et al. found that NT-3 induces the phosphorylation of TrkB receptors in cultured granule neurons, which leads to death of those neurons, in contrast to BDNF, which promotes the survival of those cells [257]. Similarly, high levels of NT-3 were reported in cerebella of individuals with autism, indicating oxidative stress-related damage to proteins [258]. Apparently, NT-3 induces dysfunction after embryonic cerebellar development, since it is associated with dysregulated calcium homeostasis [259].

In summary, NTs and NTFs play an extremely important role in regulating adult neurogenesis in various neurogenic niches throughout the brain. We suggest that the reader review the recommended articles that address the topic in-depth (see [91,210,211,216]).

## 7. The Debate on Adult Neurogenesis in Humans

One of the most discussed topics on adult neurogenesis in recent years is its relevance for human brain function and whether it persists during adulthood. The first evidence for adult neurogenesis in the human brain was presented by Eriksson et al. (1998) [3]. The authors showed evidence of newly generated cells in the hippocampus and SVZ. Tissues were derived from five postmortem brains of adults aged 57 to 72 years with cancer who received a single dose of BrdU to trace tumors. BrdU-positive cells were co-labeled with markers of granular cell neurons such as NeuN and calbindin, and this was the first report of adult neurogenesis persisting in humans throughout life. This work also made a large contribution to the field: the employment of BrdU labeling to visualize newly generated cells, which is currently the most frequently employed method to identify cell proliferation.

In 2010, Knoth et al. (2010) showed new neurons in the hippocampus from the first postnatal day up to centenary brains obtained postmortem [260]. The workgroup studied the immunolocalization of 14 markers typically associated with adult hippocampal neurogenesis in rodents with DCX-positive cells labeled in the hippocampus of the samples analyzed. The markers detected in 30 to 40 year-old tissues were Ki67, Mcm2, Sox2, nestin, Prox1, PSA-NCAM, calretinin, and NeuN. Additionally, some key markers, such as Nestin, Sox2, and Prox1, were continuously coexpressed at the oldest ages. Thus, several markers reported in murine models were also expressed in human newborn neurons, validating these cell markers for use in human tissue (see Table 7).

By 2018, the Alvarez-Buylla group, one of the most relevant research groups in the field of adult neurogenesis, published that very few newly generated cells were detected in postnatal human brain tissues, which sharply decreased with age [261]. At almost the same time, Boldrini et al. published a paper showing that healthy older subjects without cognitive impairment, neuropsychiatric disease, or treatment display preserved neurogenesis [103]. These findings started a major debate. These controversial data were discussed in a paper written by some of the most prominent leaders in the field of adult neurogenesis [167]. In these papers, researchers argued that the lack of neurogenesis presented in the study by Sorrells et al. may be due to the tissue treatment and postmortem time at which the samples were fixed, because these factors can decrease the expression of cell markers such as DCX [261]. On the other hand, Boldrini et al. clearly reported the expression of the same markers in 28 samples from adults [103]. As we argued earlier in Section 4, “The controversial new adult neurogenesis zones”, the technical question is a very important issue in animal studies. In studies with human tissue, this issue becomes even more important and critical, and might explain many of the discrepancies or differences between studies. Protocol optimization for the detection of neurogenesis is not a straightforward task (see [52]). A few months later, a new result that undoubtedly confirmed persistent neurogenesis in humans was published [4]. In this study, the authors used several combinations of fixative processes and state-of-the-art tissue processing methods. The time of the postmortem delay was well monitored, and they optimized the fixation time and avoided freezing, paraffin inclusion, or any type of mechanical alteration of the tissue. In summary, the researchers perfectly optimized the protocol for the detection of neurogenesis. They successfully tested several markers (DCX, NeuN, PSA-NCAM, GFAP, PH3, Prox1, calretinin, βIII-tubulin, and calbindin) in healthy brains, and under pathological conditions in brains from patients with Alzheimer’s disease. They identified thousands of immature neurons in the DG of 90-year-old neurologically healthy human subjects. The author reported that the neurons of healthy subjects exhibited variable degrees of maturation during the differentiation stages of adult hippocampal neurogenesis. In contrast, the number and maturation of neurons in subjects with Alzheimer’s disease progressively decreased. This very elegant and tightly controlled study confirmed the existence of adult neurogenesis in the hippocampus. An interesting direction would be to investigate nontraditional areas to establish the characteristics of these niches in the adult human brain. Determining whether new neurons are continuously incorporated into the human brain during aging is a critical question. An understanding of this process might help us to generate therapeutic strategies to prevent age-related cognitive decline, which has become a global public health problem over time.

## 8. Discussion

As we have shown in this review, adult neurogenesis is a process that most frequently occurs in niches containing adult neural precursor cells with the potential to generate new cells that eventually differentiate into neurons (see Figure 2 for a schematic summary of the most relevant markers and molecular regulators presented here). Adult neurogenesis is an exciting phenomenon that is still intriguing in the scientific community because many open questions persist, and it has been, since its first description, the source of a paradigm shift in the field. After the acceptance that adult neurogenesis occurs, only two major areas of the brain were recognized as neurogenic (the hippocampus and the SVZ). Afterward, other brain areas were controversially proposed as adult neurogenic niches. The controversial results and lack of clear and compelling evidence were likely obtained because the generation of new neurons may be very low under basal conditions and/or due to technical problems of the time of sampling/analysis. The first technique developed to detect newly generated neurons in the adult brain consisted of intracranially injecting tritiated thymidine (thymidine-^3^H) in rats and detecting cells in the cell cycle by reviewing autoradiograms [52,262]. Then, in approximately 1988, the BrdU immunohistochemistry technique was developed by Miller and Nowakowski. Since then, the i.p. injection of BrdU has been the most frequently used technique to detect new cells in the adult brain. However, low BrdU+ cell signals have been visualized in some regions of the brain. Thus, a direct injection of thymidine analogs into the brain was tested; as a result, the signal became more powerful and clear. Notably, Pérez-Martín et al. used this approach to study hypothalamic neurogenesis [104]. This example shows that many other neurogenic areas likely exist in the brain that the current techniques are not powerful enough to detect.

Another open question regarding the neurogenic zones is related to the niches. To date, major descriptive studies have been conducted in the SVZ, hippocampus, and hypothalamus. The niches share stem cell properties, such as asymmetric division, proliferation, and differentiation. All of these properties are upregulated by growth factors, and neurogenic zones are highly irrigated zones that respond to external stimuli. Although the generation of cells shares several developmental steps and has a glia-like cell as the most primitive precursor in the line, every niche has a different subset of precursor cells with a particular morphology, including the most primitive morphology. Additionally, the finding that every described niche produces a fully mature neuron according to the local circuitry is very interesting. An interesting approach would be to test whether isolated precursors change the differentiation phenotype of cells by employing specific molecular markers and factors related to the microenvironment of other niches.

Importantly, adult neurogenesis has evolutionary and survival purposes. Among them, maintaining or increasing brain and body functionality, as well as the proper deployment of the behavior of organisms, increases the chances of survival. Adult neurogenesis is modulated by different environmental demands [263]. For example, exposure to cannabinoids during the adolescent–young adulthood stage can have long-term consequences on adult neurogenesis and behavior throughout adulthood [1]. AHN has a fundamental role in learning and memory capacities [264]. Additionally, exercise, or cognitive and sexual activity, regulate the generation of new neurons in the adult hippocampus, and these new neurons exert an effect on the behavioral development of animals [263].

This review was established in an attempt to help scientists, particularly in the field of adult neurogenesis, improve their present understanding of the different neurogenic zones, the mechanisms they share, and the differences that characterize them and make them unique. The aspects that make these zones unique are potentially the key to achieving a better understanding of this fascinating phenomenon. We hope that this effort has been helpful to the scientific community and facilitates studies of adult neurogenesis.

## Figures and Tables

**Figure 1 ijms-22-11489-f001:**
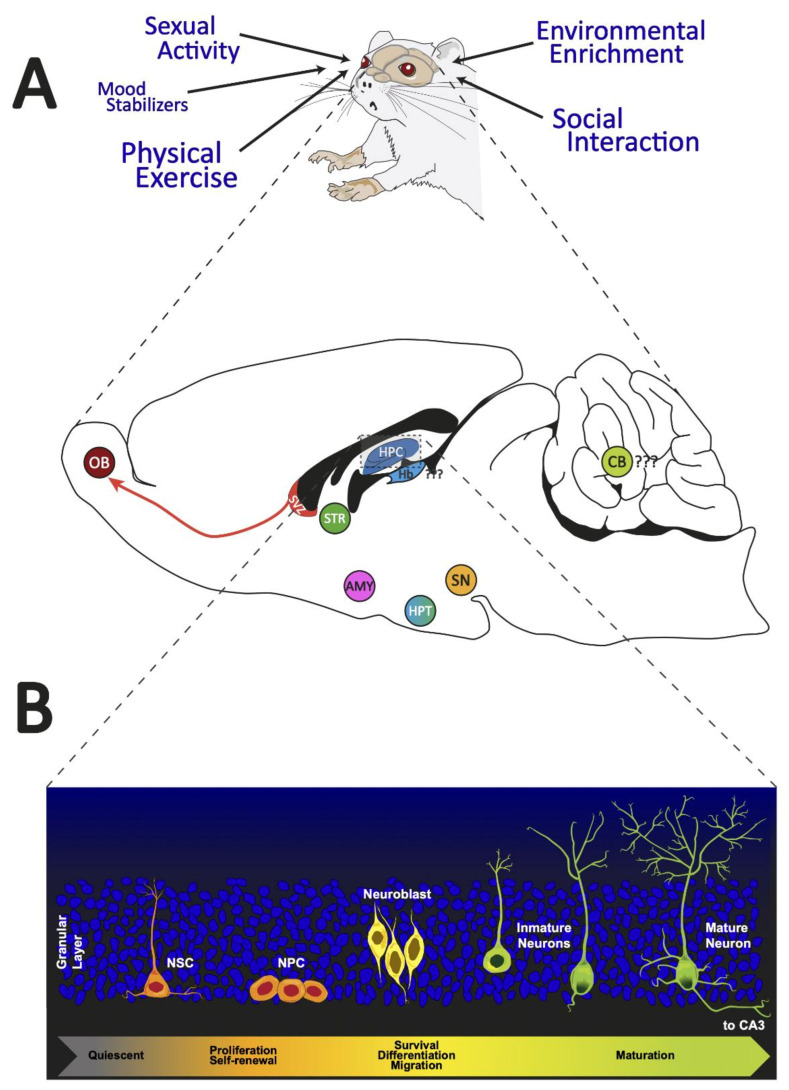
**Adult neurogenic brain zones.** Panel (**A**): We illustrated the different external regulators of adult neurogenesis. In addition, a sagittal section of the brain of a rat shows the neurogenic zones reported in adult mammals. Panel (**B**): We presented the distinct cell morphologies associated with the different stages of adult hippocampal neurogenesis. OB: olfactory bulb; SVZ: subventricular zone; STR: striatum; HPC: hippocampus; Hb: habenula; CB: cerebellum; SN: substantia nigra; HPT: hypothalamus; AMY: amygdala; NSC: neural stem cells; NPC: neural progenitor cells.

**Figure 2 ijms-22-11489-f002:**
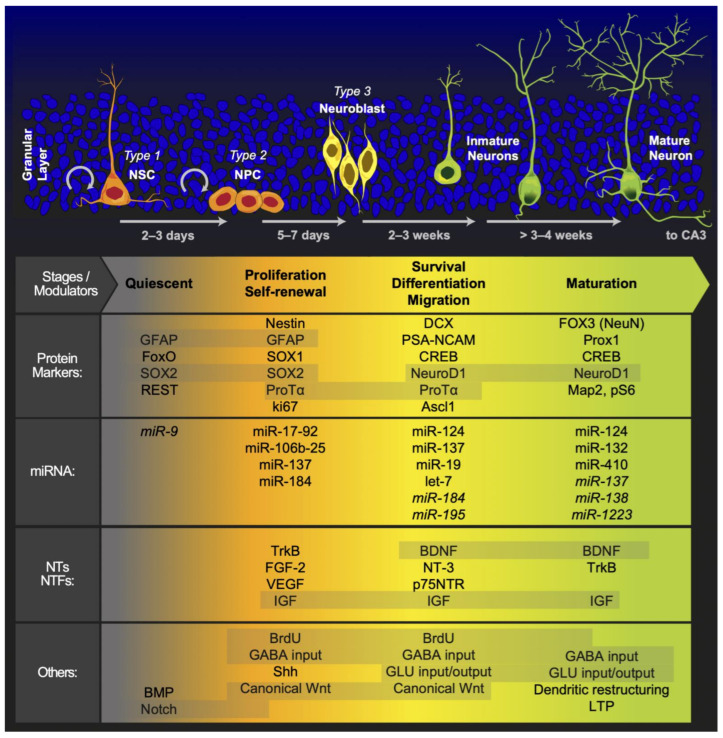
**Molecular regulators of the different stages of adult hippocampal neurogenesis.** We illustrate the distinct cell morphologies associated with the different stages of adult hippocampal neurogenesis at the top of the figure. Below, we include some of the molecular markers, NTs, NTFs, miRNAs and other factors associated with specific neurogenic stages. This list is not exhaustive; for more detailed information, please see the tables and text. In the last rows, we show the effect of stimulating the ECS. Abbreviations (not described in the main text): achaete-scute family BHLH transcription factor 1 (Ascl1), bone morphogenic proteins (BMP), microtubule-associated protein 2 (MAP2), phospho-histone 3 (pH 3), and prospero homeobox 1 (Prox1). The shadow over the molecules indicates that the process is continuous from one stage to another. The miRNAs in italics indicate that they exert an inhibitory effect on the process.

**Table 1 ijms-22-11489-t001:** Adult neurogenesis in the SVZ.

Species	Age/ Sex	Manipulation/ Treatment	Proliferation/ Differentiation	Maturation/ Survival	Protein/Gene/Growth factor	Function	Effect on the SVZ	References
Sprague-Dawley	PND 1–3	BDNF (30 ng/mL)	Sox2 NeuN		BDNF		Role in migration	[34]
		TrkB-Fc (2 μL/mL)					TrkB has roles in migration, signaling, synaptic formation, maturation and plasticity	
CDI mice	Newborn (PND 4–10) Young adult (1–2 months)	Culture medium supplemented with BDNF (50 ng/mL) or with an inhibitor.	GFAP TrkB				Mediate migratory signals	[35]
Wistar rats	Adult/ Male	SAH	Ki67 DCX GFAP				Regulation of neurogenesis after a neurological event	[36]
Qkf-GFP transgenic mice	PND 49–120	Untreated	GFAP CD24		KAT6B gene	Important role in adult neurogenesis	High expression in the SVZ	[37]
Btg1 knockout mice	PND 7– 2 months/ Male	Untreated	GFAP DCX NeuN Nestin BrdU Ki67		Btg1 gene	Cell cycle inhibitory gene	Required for the proliferation, maintenance and self-renewal of NSCs	[38]
Wistar rats	PND 1	Astrocytes isolated from the cerebral cortex Untreated	GFAP bFGF		FGF-2 Epidermal growth factor		Promotes the astrocyte hypertrophic morphology and proliferation.	[39]
Wild-type CD1 Mice	PND 30–60/ Male	Untreated	BrdU DCX		5HT and serotoninergic transmission		Critical role in proliferation. 5HT increases the proliferation of B1 cells through the activation of 5HT2Cr.	[40]
Transgenic mice	PND 14–41	Untreated	GFAP		GABA		GABA controls neuroblast proliferation through GABA_A_	[41]

5HT: serotonin; FGF-2: fibroblast growth factor-2; bFGF: basic fibroblast growth factor; SAH: subarachnoid hemorrhage.

**Table 2 ijms-22-11489-t002:** Adult neurogenesis in the hippocampus.

Species	Age/ Sex	Manipulation/ Treatment	Proliferation/ Differentiation	Maturation/ Survival	Protein/Gene/ Growth Factor	Function	Effect on AHN	References
Mice	6 weeks/ female	Experimental autoimmune encephalomyelitis	BrdU and DCX		Wnt	Wnt signaling may support neurogenic processes and immune-mediated neuroinflammation	Increases proliferation	[61]
Mice	2–3 months/ males	Excitotoxicity model	BrdU		Shh	Shh expression by mossy cells is indispensable for their survival	Increases survival	[62]
Mice	50 PND/ males	TrkB knockout	DCX	Calbindin D28K	BDNF-TrkB	BDNF-TrkB activation participates in maturation	Increases integration and maturation	[63]
C57BL/6J mice	4 weeks/ males	Kinase-dead mutant mice	Ki-67 and DCX		MSK1	MSK1 does not alter the basal rate of proliferation MSK1 negatively regulates the number of cells destined to become neurons	No effect on the basal proliferation rate	[64]
C57BL/6N mice	8 weeks/ males	Alzheimer’s disease model and physical exercise	DCX		MAPK	Exercise prevents AD MAPK regulates AHN	Increases proliferation	[65]
C57BL/6 mice	3–6 months/ sex not specified	Phosphorylation-competent p300 (G442S) knock-in (KI) mouse model	BrdU and Ki-67	NeuN	Phosphorylation- competent p300	Changes in p300 phosphorylation modulate AHN	Increases cell survival	[66]
C57BL/6 mice	10–15 weeks/ male	ProTα+/− knockout mice	BrdU		Downregulated genes: Nrp1, Racgap1, Nrxn3, and DCX	ProTα+/− impairs learning and memory, and hypolocomotor activity. Genes are related to anxiety, learning/memory-functions. ProTα+/− mice: AHN was downregulated	Decreases proliferation and survival	[67]

Note: MSK1: mitogen- and stress-activated protein kinase 1.

**Table 3 ijms-22-11489-t003:** Adult neurogenesis in the hypothalamus.

Species	Age/ Sex	Manipulation/ Treatment	Proliferation/ Differentiation	Maturation/ Survival	Protein/Gene/ Growth Factor	Function	Effect on the Hypothalamus	References
Rat	3 months/males	Untreated	BrdU and DXC	NeuN and DARPP-3	Agouti-related peptide	Orexigenic agent	Increases proliferation	[23]
C57Bl/6 and CD-1 mice	P21 and P42 males and females	Untreated			Notch 1 and 2	Crucial pathway to maintain NSC behavior	These proteins are expressed in the niche	[115]
C57BL/6	3-month-old	Chronic high-fat-diet feeding	BrdU	Tuj1/NeuN	IKKβ/NF-κB	Controls cell survival, growth, apoptosis and differentiation	Activated when neurogenesis is inhibited	[116]
C57BL/6	3-month-old	Chronic high-fat-diet feeding	BrdU	Tuj1/NeuN	Sox 2	Involved in pluripotency	It is expressed in neurospheres derived from the hypothalamus	[116]
C57BL/6	3-month-old	Chronic high-fat-diet feeding	BrdU	Tuj1/NeuN	ARC	Intermediary gene expressed in cells with the capacity of firing	It is expressed in neurospheres derived from the hypothalamus	[116]
C57Bl/6 and CD-1 mice	P21 and P42 males and females	Untreated			Sox 9	Is a crucial factor for the induction of proliferation and maintenance of the neurogenic pool	It is expressed in the niche	[115]
C57Bl/6 and CD-1 mice	P21 and P42 males and females	Untreated			Hes 1	It is a protein that controls the proper timing of neurogenesis and morphogenesis	It is expressed in the niche	[115]
C57Bl/6 and CD-1 mice	P21 and P42 males and females	Untreatment			CD63	Participates in modulating the formation of new neurons	It is expressed in the niche	[115]
C57Bl/6 and CD-1 mice	P21 and P42 males and females	Untreated			FZD5	Plays a key role in regulating the cell fate commitment	It is expressed in the niche	[115]
C57Bl/6 and CD-1 mice	P21 and P42 males and females	Untreated			NTrk-2T1	Surface protein involved in proliferation	It is expressed in the niche	[115]
C57Bl/6 and CD-1 mice	P21 and P42 males and females	Untreated			Thrsp	Thyroid hormone-responsive gene	It is expressed in the niche	[115]
Mice	P19	High-fat diet	BrdU i.p. Nestin		Hu	Progenitor cells marker	Expressed in tanycytes	[107]
Wistar Rats	2 months old	Microdoses of IGF-I administered with minipumps	BrdU	vimentin	IGF-I	Insulin-like growth factor I (IGF-I).	Participates in proliferation, differentiation and survival	[104]

**Table 4 ijms-22-11489-t004:** Adult neurogenesis in the SN.

Species	Age/ Sex	Manipulation/ Treatment	Proliferation/ Differentiation	Maturation/ Survival	Protein/Gene/ Growth Factor	Function	Effect on the SN	References
Mice	2–20 months/males	BrdU	BrdU, tyrosine hydroxylase (TH), nestin, fluorogold	NeuN, CRMP-4	Stem cells	Differentiation into new tyrosine hydroxylase-positive (dopaminergic) neurons	None	[26]
Mice and rats	10 weeks/females (mice), males (rats)	Untreated	Polysialic acid, TH, NG2, GFP, BrdU	GFAP, GSTP1	Undifferentiated cells	Differentiation into glial cells, especially microglia	Increases differentiation	[126]
Rats	Not reported/females	BrdU, dopamine D3 receptor agonist: 7-OH-DPAT	BrdU, PCNA, TH	GFAP, NeuN	Undifferentiated cells	Neurogenesis and neuronal differentiation into the dopaminergic phenotype	Increases differentiation	[127]
Mice	Not reported	Untreated/Transgenic	TH, ChAT, GAD	GDNF, NeuN, Parv	Sonic Hedgehog (SHh)	Maintaining homeostasis through a noncell autonomous process; also involved in cellular differentiation, maintenance and survival	Promotes differentiation, maintenance and survival	[128,129]

**Table 5 ijms-22-11489-t005:** Adult neurogenesis in the striatum.

Species	Age/Sex	Manipulation/Treatment	Proliferation/ Differentiation	Maturation/ Survival	Protein/Gene/ Growth Factor	Function	Effect on the Striatum	References
Sprague-Dawley rats	9–10 weeks of age/males	Untreated	BrdU	DCX, DCX/NeuN, DCX/CRMP4, GAD-67,GABA, and CR		Markers of the progenitors and migration of cells and interneurons	Increases proliferation and migration and the number of specific interneuron classes	[137]
G*tv-a* and CAG-CAT-EGFP Transgenic Mice	9–16 weeks of age/males	MCAO	DCX, Tuj-1	NeuN, GFP, GFAP GST-π PECAM-1 Cre-*lox*P		Progenitor and migration markers	Increases proliferation and migration	[139]
Wistar rats	8–10 weeks of age/males	MCAO and GFP injection	BrdU	DCX DCX/BrdU NeuN DARPP-32		Progenitor and migration markers	Increases proliferation and migration	[140]
Sprague-Dawley rats	Adult/males	6-OHDA lesion and an infusion of TGF-α	BrdU	DCX Β-III tubulin		Progenitor and migration markers	Substantial induction of proliferation, migration, and differentiation	[27]
Wistar rats	Adult/males	Quinolinic acid (QA) lesion	BrdU DCX	DCX/NeuN		Progenitor and migration markers	Increases proliferation and migration	[141]
Macaque Monkeys (*Macaca fuscata*)	5–11 years/females	MCAO	BrdU Musashi 1 Nestin	βIII-Tubulin NeuN	Tbr1 Islet1	Progenitor and migration markers	Increases proliferation and migration	[142]
Squirrel Monkeys (*Saimiri sciureus*)	4–6 years of age/males	Enriched environment	BrdU	NeuN		Progenitor and migration markers	Increases proliferation and migration	[143]
Rabbits (*Orictolagus cuniculus*)	Adult/females	Untreated	DCX/BrdU	BrdU/NeuN BrdU/Calretinin DCX/PSA-NCAM β-Tubulin HuC/D protein		Progenitor and migration markers	Increases proliferation and migration. Localization of neuronal precursors	[144]
Human	21 to 68 years of age	Untreatment	DCX/PSA-NCAM	DCX/NeuN NeuN/SOX10 Neun/DARPP-32		Progenitor and migration markers	Increases proliferation and migration.	[145]
Athymic NIH FOXN1-RNU Rats	Adult/males	QA lesion and graft of stem cell-derived human striatal progenitors		DARPP-32/CTIP2 GABA Calbindin (CB) CR GFAP NESTIN		Migration markers	Migration of medium spiny neurons in humans	[147]
B6C3-Tg (APPswe, PSENIde9) 85Dbo/J Transgenic mice	3 mouths of age/males	Plasmid infusion		NeuN Nestin DCX GFAP		Progenitor and migration markers	Increases proliferation and migration	[149]
Cultured human dermal fibroblasts	Adult	Lentiviral vector injection (miRNA-9/9*-124, Bcl-xL, Dox (doxicline) and EF1α		MAP2 β-III tubulin GABA GAD67 FOXP1 DLX5 DARPP-32 NeuN	BCl11B (CT1P2), DLX1, DLX2 and MYT1 L	Markers of progenitor cells and interneurons markers	Promote neuronal differentiation and survival, as well as the formation of specific interneuron class	[148]
Wistar rats	3–4 months/male	MCAO Transfection of miR-124a in vitro	BrdU DCX-GFP		↓JAGI Inactivation of Notch pathway ↑p27Kip1	Progenitor cells markers	Reduce NPC proliferation and promote neuronal differentiation in the SVZ	[155]
C57BL/6J mice	3–4 months/males	MCAO and injection of a lentivirus (miR17-92 cluster)	BrdU, Tuj1, NG2	GFP	PTEN ENH1	Progenitor cell markers	Proliferation and survival of neuronal progenitor cells in the SVZ	[156]
Sprague-Dawley rats	Adult/males	BDNF infusion	BrdU MAP-2	TuJ1 GFPA	BDNF	Progenitor cells markers	Increased proliferation	[157]

MCAO, middle cerebral artery occlusion; DCX, doblecortin; BrdU, bromodeoxyuridine; NeuN, neuronal nuclear; CR+, calretinin; PSA-NCAM, polysialylated neuronal cell adhesion molecule; TGF-α, transforming growth factor-α; Hu; TNFR-1, tumor necrosis factor receptor 1; SDF-1, stromal cell-derived factor 1; MCP-1, monocyte chemoattractant protein 1; Tuj-1, βIII-tubulin; PECAM-1, platelet/endothelial cell adhesion molecule-1; GST-π, glutathione *S*-transferase; DARPP-32, dopamine and adenosine 3’-5´monophosphate-regulated phosphoprotein with a molecular weight of 32 kD; GFP, green fluorescent protein; JAG1, Jagger-1.

**Table 6 ijms-22-11489-t006:** Adult neurogenesis in the cerebellum and habenula.

Species	Age/ Sex	Manipulation/ Treatment	Proliferation/ Differentiation	Maturation/ Survival	Protein/Gene/Growth Factor	Function	Effect on Specific Zones	References
Mice		Transgenic: CD1	β—III tubulin		Orthodenticle homeobox 2 (Oxt2)	Transcription factor	Regulates the activity of other genes	[168]
Mice		Transgenic: Catnblox(ex3)/+ Apclox/lox Catnblox(ex2-6)	Ki67		Wnt-1	Signaling protein	Promotes the proliferation of NSCs	[169]
New Zealand White Rabbits	2–5 months, and 1–3 years	No-treatment	BrdU, Pax2, Sox2, Olig2	NeuN, Pax6, Pax2, Sox2, Olig2	PSA-NCAM+ precursors, Map5+ cells	Developmental markers	Glial and neuronal progenitors	[170]
Mice 129xMF1	8 weeks	Transgenic mice In situ hybridization	Calbindin, BLBP (brain lipid-binding protein)		Sox1 Sox 2	Regulators of the self-renewal and differentiation of neuronal progenitors	Astroglial cell type development (Bergmann glia)	[171]
Human	Adult		Calbindin GFAP		Sox1 Sox2 Sox9	Regulators of the self-renewal and differentiation of neuronal progenitors	Astroglial cell type development (Bergmann glia)	[172]
Rats	E18-P2	Dissociated and organotypic cultures	Calbindin, NSE NG2	GFAP	NGF (survival) BNDF, NT-3 (immature cells)		Differentiation of cerebellar neurons	[173]
Mouse C57BL/6J	5 days	CGC cultures	Primers: 5-LOX, DNMT1, DNMT3a, cyclophilin mRNA		5-LOX	Key enzyme in the biosynthesis of the inflammatory leukotrienes and anti-inflammatory lipoxins	Regulation of neural stem cells, proliferation and differentiation. Increases in aging	[174]
Mice	8 weeks/males	Fluoxetine (in drinking water; 155 mg/L for 4 weeks)	BrdU BDNF nestin+ NPCs		Increased expression of the BDNF mRNA	Buffering stress responses and in mediating behavioral responses	BDNF promotes cell proliferation and neurogenesis	[29]
Rats	NS/ adults	BDNF infusion in the lateral ventricle	BrdU TrkB		TrkB levels increased	TrkB expression correlates with the level of BrdU expression	TrkB, a receptor for BDNF, mediates cell proliferation in the habenula	[157]

Note: Shaded rows indicate evidence of adult neurogenesis in mammals in the habenula, light rows indicate evidence in the cerebellum.

**Table 7 ijms-22-11489-t007:** Summary of adult human neurogenesis studies.

Age (Years)	Sample Number	Source	Tissue Preparation	Proliferation Marker	Cell Fate Markers	References
57–72	5	Postmortem tissue	24 h postfixed with 4% paraformaldehyde and then transferred to a 30% sucrose solution	BrdU	NeuN Calbindin GFAP	[3]
1 day–100 years	3 fetal 49 brains 1 resection	Postmortem tissue Postmortem tissue Lobectomy	Paraffin sections Not described Not described	PCNA Ki67	DCX Nestin BLBP NeuroD Prox 1 Sox2 PSN-NCAM Calretinin Calbindin GAD65 GAD67 TUC4 β-III-tubulin Map2ab	[260]
14–79	28 brains	Postmortem tissue	Coronal blocks were flash-frozen in liquid Freon (−20 C°) and stored at −80 °C. The tissue samples were fixed with formalin. For processing, the hippocampus was dissected from the blocks, fixed with 4% paraformaldehyde at 4 °C and then cryoprotected in 30% sucrose	Ki-67	Nestin DCX PSA-NCAM Sox2	[103]
18–77	17 brains 12 surgically resected tissues from patients with epilepsy as controls	Postmortem tissue Lobectomy		Ki-67	DCX PSA-NCAM NeuN	[261]
43–87 40–100	17 healthy subjects to establish neurogenesis 13 healthy control subjects 45 patients with AD	Postmortem tissue postmortem tissue	Tissues were stored in 4% PFA at 4 °C for 24 h Tissues were stored in 4% PFA at 4 °C for 24 h		DCX NeuN Prox1 PH3 Calretinin Calbindin DCX	[4]

## Data Availability

Not applicable.

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
