# Peer review of "Adult Neurogenesis: A Story Ranging from Controversial New Neurogenic Areas and Human Adult Neurogenesis to Molecular Regulation"

_ijms, 2021, doi:10.3390/ijms222111489_

Round 1

Reviewer 1 Report

The paper by Leal-Galicia et al. entitled “Adult neurogenesis: a story beyond subventricular zone and hippocampus”  is a review article. A key goal of this review manuscript is to describe structures in the adult brain that are involved in generation of new neurons, including the two neurogenic structures, the subventricular zone of the lateral ventricle and the dentate gyrus of the hippocampal formation. After first works Altman (Altman 1962, 1963; but not Altman and Das, 1965 as referred by the authors), the two neurogenic structures were intensively studied in mammalian adult brain. Over time, newly generated neurons were found in other brain structures such as the cerebral cortex, the striatum, the substantia nigra, the amygdala and the hypothalamus. However, the origin of these new neurons is still debate. Several studies report that new neurons are generated in the SVZ, and then migrate to these brain structures while others show that they are generated within these structures.

I assumed the submitted manuscript should be an updated review article. However, I found a similar review paper by Jurkowski et al. published a year ago. Interestingly, the authors refer this article in their manuscript. I wonder, whether the authors have made a close comparison of their manuscript with that published review paper. The submitted manuscript contains very similar sections (2.1., 2.2., 3.2., 3.3. and 3.4.) and I would like to know how and in what aspects these sections are differ from already published review by Jurkowski et al. In order not to be similar, the content of the manuscripts needs to be completely reorganized. I think the authors should modify these sections putting more emphasis on functions of both hippocampal and SVZ adult neurogenesis in mammals. Maybe it will be better among others to discuss more controversial data on adult neurogenesis in humans, or to discuss the effects of neurotrophins on adult hippocampal neurogenesis highlighting neurotrophin-3 instead of BDNF that has already been described in the paper by Jurkowski et al. The title should also be changed as it is very similar to the published review paper. It contains the same words in a different order. Please compare “Beyond the Hippocampus and the SVZ: Adult Neurogenesis Throughout the Brain” by Jurkowski et al vs “Adult neurogenesis: a story beyond subventricular zone and hippocampus”.   

I have also a major concern about the remaining two sections, 3.1 (Adult Neurogenesis in the habenula. Mechanism and possible functional implications) and  3.5. (Adult Neurogenesis in the Cerebellum. Mechanism and possible functional implications) that are not matched with the topic and therefore cannot be included in the submitted manuscript.

The section 3.1 is focused on adult neurogenesis in the habenula. I believe that this submitted review paper is about adult neurogenesis in mammals. If so, why a majority of data concerning adult neurogenesis in the habenula relates to fishes. The authors considered only 2 papers on adult neurogenesis in the habenula of mammalian species. The paper by Sachs and Caron (2015) demonstrates that chronic treatment with fluoxetine enhances cell proliferation in the medial habenula of Nestin-Cre-ER-Tomato (NCerT) mice.  In this research, neurogenesis occurs in the habenula of  transgenic mice but not of wild type mice. The second paper by Pencea et al. (2001) reports the presence of new neurons in the habenula of BDNF treated mice. Thus, BDNF  promotes cell proliferation in the habenula of adult brain. Therefore, the question should be addressed whether adult neurogenesis occurs in the habenula of mice under normal physiological conditions.

The section 3.5 describes adult neurogenesis in the cerebellum. This process is not called adult neurogenesis. In eutherian mammals, most neurogenesis is completed embryonically. However,  the cerebellum is one of the late developing brain structure. In rodents, specifically in rat, neurons of the deep nuclei and Purkinje cells of the cerebellum are generated during embryonic development on day E14 and E15, while the generation and development of cerebellar interneurons and granule cells occurs postnatally, after birth. Postnatal generation of neurons in the cerebellum lasts for over 3 weeks in rats. In humans, it takes place during the first year of life. This process is known as developmental neurogenesis that is a well-preserved feature of marsupials. In marsupials, many brain structures develop after birth  including the cerebral cortex.

Author Response

General Statements about the Revised Manuscript

We strived to cover most of the reviewers' comments. We addressed those comments that we felt were the most relevant and those who stuck to our article's focus. In other cases, we justify the reasons for noncompliance in the Point-by-Point Response to Comments section of this letter (see below). We also performed minor modifications throughout the article adds to those kindly recommended by reviewers to improve clarity and accuracy. Changes in the manuscript were agreed on by all authors. All revisions/modifications of the manuscript were highlighted using a yellow marker for easy identification. An academic English editor service was hired to revise and proofread the manuscript. All the changes in the manuscript by both the editor and us were highlighted too. You will find below a point-by-point response to your' comments. Regular font style is used for featured comments, and italics have been used to show our responses.

Point-by-Point Response to Reviewer Comments

EDITORAL COMMENTS

RESPONSE: We thank the reviewer for the comments made about our article. We have made several and deep changes to our manuscript. We rewrote all the paragraphs of section 2.2 "Adult Neurogenesis in the hippocampus. Mechanism and possible functional implications", which had few sentences are almost the same with the published papers. They are highlighted with yellow color.

REVIEWER 1:

The paper by Leal-Galicia et al. entitled “Adult neurogenesis: a story beyond subventricular zone and hippocampus”  is a review article. A key goal of this review manuscript is to describe structures in the adult brain that are involved in generation of new neurons, including the two neurogenic structures, the subventricular zone of the lateral ventricle and the dentate gyrus of the hippocampal formation. After first works Altman (Altman 1962, 1963; but not Altman and Das, 1965 as referred by the authors), the two neurogenic structures were intensively studied in mammalian adult brain. Over time, newly generated neurons were found in other brain structures such as the cerebral cortex, the striatum, the substantia nigra, the amygdala and the hypothalamus.  However, the origin of these new neurons is still debate. Several studies report that new neurons are generated in the SVZ, and then migrate to these brain structures while others show that they are generated within these structures.

  1. I assumed the submitted manuscript should be an updated review article. However, I found a similar review paper by Jurkowski et al. published a year ago. Interestingly, the authors refer this article in their manuscript. I wonder, whether the authors have made a close comparison of their manuscript with that published review paper. The submitted manuscript contains very similar sections (2.1., 2.2., 3.2., 3.3. and 3.4.) and I would like to know how and in what aspects these sections are differ from already published review by Jurkowski et al. In order not to be similar, the content of the manuscripts needs to be completely reorganized. I think the authors should modify these sections putting more emphasis on functions of both hippocampal and SVZ adult neurogenesis in mammals. Maybe it will be better among others to discuss more controversial data on adult neurogenesis in humans, or to discuss the effects of neurotrophins on adult hippocampal neurogenesis highlighting neurotrophin-3 instead of BDNF that has already been described in the paper by Jurkowski et al.

RESPONSE: We thank the reviewer for the comments made about our article. We have made several and deep changes to our manuscript, as suggested. We have changed the focus of the manuscript by concentrating heavily on criticism and debates around the new neurogenic niches described, particularly those with less evidence, such as the habenula and the cerebellum. In that regard, we have added evidence of the presence of mammalian adult neurogenesis in both structures, and extensively discuss the evidence and future perspectives. We have added several sections to the manuscript where we evidenced the participation of molecular regulators in the adult neurogenic process, such as miRNA, neurotrophic factors, and neurotrophins, which were widely discussed. Likewise, we have added a section to discuss the evidence on adult neurogenesis in humans, as well as the debate that has arisen in the face of contrary evidence. Also, we have added seven tables and a figure that summarizes the information.

  1. The title should also be changed as it is very similar to the published review paper. It contains the same words in a different order.

RESPONSE: We thank the reviewer for the comments made about our article. We have made several and deep changes to our manuscript, which have led to the change of its name, as suggested. The new name is "Adult neurogenesis: a story from controversial new neurogenic areas and human adult neurogenesis to the molecular regulation", which we believe better reflects what is contained in the manuscript.

  1. Please compare “Beyond the Hippocampus and the SVZ: Adult Neurogenesis Throughout the Brain” by Jurkowski et al vs “Adult neurogenesis: a story beyond subventricular zone and hippocampus”.

RESPONSE: A previous review discussing the different neurogenic niches was carried out by Jurkowski et al (). The group lead by Gil-Mohapel, carried out an interesting revision on the different neurogenic zones, their functionality and offers a complete comparison among the species studied to characterize the neurogenesis, the protocols employed to label cell proliferation, differentiation, and survival as well as the different treatments performed in the studies that are discussed in the paper. To this information, we additionally offer a comparison among the niches as well as, include tables showing the different proteins, growth factors, and genes involved in the neurogenesis process in the proliferative regions. To enrich the information, we include less studied zones in the postnatal neurogenesis: cerebellum and habenula. In the current review, we revise the debate on the persistence of neurogenesis in the human adult brain, which was a matter of discussion in recent years. Both works, Jurkowski’s and ours, offer a comprehensive review on the knowledge regarding the adult neurogenic areas with different information that can be complementary for the readers interested in obtaining a complete view of the phenomenon.

  1. I have also a major concern about the remaining two sections, 3.1 (Adult Neurogenesis in the habenula. Mechanism and possible functional implications) and 5. (Adult Neurogenesis in the Cerebellum. Mechanism and possible functional implications) that are not matched with the topic and therefore cannot be included in the submitted manuscript. The section 3.1 is focused on adult neurogenesis in the habenula. I believe that this submitted review paper is about adult neurogenesis in mammals. If so, why a majority of data concerning adult neurogenesis in the habenula relates to fishes. The authors considered only 2 papers on adult neurogenesis in the habenula of mammalian species. The paper by Sachs and Caron (2015) demonstrates that chronic treatment with fluoxetine enhances cell proliferation in the medial habenula of Nestin-Cre-ER-Tomato (NCerT) mice.  In this research, neurogenesis occurs in the habenula of  transgenic mice but not of wild type mice. The second paper by Pencea et al. (2001) reports the presence of new neurons in the habenula of BDNF treated mice. Thus, BDNF  promotes cell proliferation in the habenula of adult brain. Therefore, the question should be addressed whether adult neurogenesis occurs in the habenula of mice under normal physiological conditions.

RESPONSE: We thank the reviewer for the comments made about our article. We have changed the focus of the manuscript by concentrating heavily on criticism and debates around the new neurogenic niches described, particularly those with less evidence, such as the habenula. In that regard, we have added evidence and a very extended discussion about the presence of mammalian adult neurogenesis in this structure. Please see section 4. “The controversial new adult neurogenesis zones”, page 23; particularly section 4.1 “Adult neurogenesis in the habenula. Mechanism and possible functional implications” in page 25.

  1. The section 3.5 describes adult neurogenesis in the cerebellum. This process is not called adult neurogenesis. In eutherian mammals, most neurogenesis is completed embryonically. However, the cerebellum is one of the late developing brain structure. In rodents, specifically in rat, neurons of the deep nuclei and Purkinje cells of the cerebellum are generated during embryonic development on day E14 and E15, while the generation and development of cerebellar interneurons and granule cells occurs postnatally, after birth. Postnatal generation of neurons in the cerebellum lasts for over 3 weeks in rats. In humans, it takes place during the first year of life. This process is known as developmental neurogenesis that is a well-preserved feature of marsupials. In marsupials, many brain structures develop after birth  including the cerebral cortex.

RESPONSE: We thank the reviewer for the comments made about our article. We have changed the focus of the manuscript by concentrating heavily on criticism and debates around the new neurogenic niches described, particularly those with less evidence, such as the habenula and the cerebellum. In that regard, we have added evidence of the presence of mammalian adult neurogenesis in the cerebellum, and extensively discuss the evidence and future perspectives. Please see section 4. “The controversial new adult neurogenesis zones”, page 23; particularly section 4.2  “Adult neurogenesis in the cerebellum. Mechanism and possible functional implications” in page 26.

Reviewer 2 Report

The manuscript reviewed the advancement of adult neurogenesis in the traditional regions including SVZ and the SGZ of hippocampus, and new regions, such as the habenula, Hypothalamus, Substantia Nigra, Striatum. The paper will provide a better understanding of the functions of adult neurogenesis under physiological and disease conditions. There are the two major concerns. First, the progress of mechanisms underlying adult neurogenesis is incomplete. For instance, the functional roles of microRNAs and epigenetic factors in the adult neurogenesis were not laid out and highlighted. Second, the roles of genes in the adult neurogenesis and related references need to be listed in the tables to allow the reader to read and get access fast.

Author Response

General Statements about the Revised Manuscript

We strived to cover most of the reviewers' comments. We addressed those comments that we felt were the most relevant and those who stuck to our article's focus. In other cases, we justify the reasons for noncompliance in the Point-by-Point Response to Comments section of this letter (see below). We also performed minor modifications throughout the article adds to those kindly recommended by reviewers to improve clarity and accuracy. Changes in the manuscript were agreed on by all authors. All revisions/modifications of the manuscript were highlighted using a yellow marker for easy identification. An academic English editor service was hired to revise and proofread the manuscript. All the changes in the manuscript by both the editor and us were highlighted too. You will find below a point-by-point response to your' comments. Regular font style is used for featured comments, and italics have been used to show our responses.

Point-by-Point Response to Reviewer Comments

REVIEWER 2:

Comments and Suggestions for Authors

The manuscript reviewed the advancement of adult neurogenesis in the traditional regions including SVZ and the SGZ of hippocampus, and new regions, such as the habenula, Hypothalamus, Substantia Nigra, Striatum. The paper will provide a better understanding of the functions of adult neurogenesis under physiological and disease conditions.

There are the two major concerns.

  1. First, the progress of mechanisms underlying adult neurogenesis is incomplete. For instance, the functional roles of microRNAs and epigenetic factors in the adult neurogenesis were not laid out and highlighted.

RESPONSE: We thank the reviewer for the comments made about our article. We have made several and deep changes to our manuscript, as suggested. We have added several sections to the manuscript where we evidenced the participation of molecular regulators in the adult neurogenic process, such as miRNA, neurotrophic factors, and neurotrophins, which were widely discussed.

  1. Second, the roles of genes in the adult neurogenesis and related references need to be listed in the tables to allow the reader to read and get access fast.

RESPONSE: We thank the reviewer for the comments made about our article. We have made several and deep changes to our manuscript, as suggested. We have added seven tables and a figure that summarizes the information.

Round 2

Reviewer 1 Report

The authors have clearly improved the manuscript. However, a few minor issues, typos and language should be revised.

  1. BrdU is not a neuronal marker, BrdU is a proliferation marker (a marker for dividing cells), and this needs to be corrected throughout the text. Examples:

Line 557: “These new neurons expressed markers of developing: bromodeoxyuridine (BrdU)”

Line 594: “this cells express early neuronal markets”

replace “this cells” with “these cells”

replace “markets” with “markers”

The word “market/s” appears in: L560, L579, L595, L612, L685-L686 (Table 5) etc.

  1. The cerebellum contains the cerebellar deep nuclei and the cerebellar cortex which consists of three layers: the outer molecular layer, the middle Purkinje cell layer and the inner granular layer, that is the deepest layer of the cerebellum.

Line 869: “cerebellum is tightly organized as a trilaminar structure consisting of an outer granular cell layer, a middle Purkinje cell layer, and an inner molecular cell layer [176]”. This sentence needs to be corrected.

  1. Bayer et al. (1995) reported that in man the Purkinje cells and the deep cerebellar nuclei are generated from the 5th (35 days) to the 6th (42 days) weeks of development.

Line 877-878: “The development of the cerebellum starts at embryonic stages and continues postnatally; in humans, this development extends from 30 postnatal days until the second year [181].” Replace “30 postnatal days” with “35-42 embryonic days”

  1. An abbreviation needs to be defined only once in the text.

Line 543: “precursors of the subventricular zone (SVZ),”. This SVZ abbreviation has already been defined (L.60).

Line 1075: nerve growth factor (NGF); and Line 1210:  Nerve Growth Factor (NGF)

Line 1076: brain-derived neurotrophic factor (BDNF), and Line 1211: Brain-derived neurotrophic factor (BNDF)

and etc. This needs to be corrected throughout the text.

Line 148: “.cThe main growth factors” remove “c”

Line 680: “see Table 5 thats shows” replace “thats” with “that”

Line 655: “others molecular mechanism poorly understood” I think it should be “other molecular mechanisms poorly understood”

Line 749: replace “Kempermann et al., 2018” with “Kempermann et al., (2018)”,

Line 1145: replace “Prominent 1” with Prominin-1

Line 1196: replace “por example,” with “for example”

Please check grammar:

Line 471: “that neurogenesis in the SN result in an increase in dopaminergic”

Line 662: “through the inhibition of phosphorylation of AKT that induce a reduction in the”

Line 667-668:  “The blockaded of TNF-R1 signaling might promote the proliferation of cells in the SVZ and neuroblast formation were enhanced [140]”

Line 708: “that the adult neurogenesis in the amygdala response to hormones.”

Line 1184: “In the striatum have been demonstrated”

and etc.

Line 722: “Female C57BL6/J living 40 under environmental enrichment showed an increase in the expression of proteoglycan neuron-glia 2 (NG2) a marker that represents parenchymal precursor cells.” The citation is missing.

Author Response

5 October 2021

Estela Castilla Ortega, PhD

Guest Editor

Dear Dr. Castilla Ortega,

Subject: “Adult neurogenesis: a story from controversial new neurogenic areas and human adult neurogenesis to the molecular regulation”, now titled “Adult neurogenesis: a story ranging from controversial new neurogenic areas and human adult neurogenesis to molecular regulation”. Manuscript No. ijms- 1378062.

Thank you for your email, enclosing the editor's comments. We have carefully reviewed the comments and have revised the manuscript accordingly. Our responses are given in a point-by-point manner below. The changes to the manuscript have been highlighted to identify all the major manuscript edits. The corrections and suggestions provided by you helped improve the paper. We hope the revised version is now suitable for publication, and we look forward to hearing from you in due course.

Sincerely,

Mario Buenrostro-Jauregui, Ph. D.

Universidad Iberoamericana

México City

México

[email protected]

General Statements about the Revised Manuscript

We strived to cover most of the reviewers' comments. We addressed those comments that we felt were the most relevant and those who stuck to our article's focus. In other cases, we justify the reasons for noncompliance in the Point-by-Point Response to Comments section of this letter (see below). We also performed minor modifications throughout the article adds to those kindly recommended by reviewers to improve clarity and accuracy. Changes in the manuscript were agreed on by all authors. All revisions/modifications of the manuscript were highlighted using a the MS Word’s “Track Changes” for easy identification. An academic English editor service was hired to revise and proofread the manuscript. All the changes in the manuscript by both the editor and us were highlighted too. You will find below a point-by-point response to your' comments. Regular font style is used for featured comments, and italics have been used to show our responses.

Point-by-Point Response to Reviewer Comments

REVIEWER 1:

The authors have clearly improved the manuscript. However, a few minor issues, typos and language should be revised.

BrdU is not a neuronal marker, BrdU is a proliferation marker (a marker for dividing cells), and this needs to be corrected throughout the text. Examples:

Line 557: “These new neurons expressed markers of developing: bromodeoxyuridine (BrdU)”

RESPONSE: This issue was solved.

Line 594: “this cells express early neuronal markets”

RESPONSE: This issue was solved.

replace “this cells” with “these cells”

RESPONSE: This issue was solved.

replace “markets” with “markers”

RESPONSE: This issue was solved.

The word “market/s” appears in: L560, L579, L595, L612, L685-L686 (Table 5) etc.

RESPONSE: This issue was solved.

The cerebellum contains the cerebellar deep nuclei and the cerebellar cortex which consists of three layers: the outer molecular layer, the middle Purkinje cell layer and the inner granular layer, that is the deepest layer of the cerebellum.

Line 869: “cerebellum is tightly organized as a trilaminar structure consisting of an outer granular cell layer, a middle Purkinje cell layer, and an inner molecular cell layer [176]”. This sentence needs to be corrected.

RESPONSE: This issue was solved.

Bayer et al. (1995) reported that in man the Purkinje cells and the deep cerebellar nuclei are generated from the 5th (35 days) to the 6th (42 days) weeks of development.

Line 877-878: “The development of the cerebellum starts at embryonic stages and continues postnatally; in humans, this development extends from 30 postnatal days until the second year [181].” Replace “30 postnatal days” with “35-42 embryonic days”

RESPONSE: This issue was solved.

An abbreviation needs to be defined only once in the text.

Line 543: “precursors of the subventricular zone (SVZ),”. This SVZ abbreviation has already been defined (L.60).

RESPONSE: This issue was solved.

Line 1075: nerve growth factor (NGF); and Line 1210:  Nerve Growth Factor (NGF)

RESPONSE: This issue was solved.

Line 1076: brain-derived neurotrophic factor (BDNF), and Line 1211: Brain-derived neurotrophic factor (BNDF)

RESPONSE: This issue was solved.

and etc. This needs to be corrected throughout the text.

RESPONSE: This issue was solved. We checked and corrected all abbreviations.

Line 148: “.cThe main growth factors” remove “c”

RESPONSE: This issue was solved.

Line 680: “see Table 5 thats shows” replace “thats” with “that”

RESPONSE: This issue was solved.

Line 655: “others molecular mechanism poorly understood” I think it should be “other molecular mechanisms poorly understood”

RESPONSE: This issue was solved.

Line 749: replace “Kempermann et al., 2018” with “Kempermann et al., (2018)”,

RESPONSE: This issue was solved.

Line 1145: replace “Prominent 1” with Prominin-1

RESPONSE: This issue was solved.

Line 1196: replace “por example,” with “for example”

RESPONSE: This issue was solved.

Please check grammar:

Line 471: “that neurogenesis in the SN result in an increase in dopaminergic”

RESPONSE: This issue was solved.

Line 662: “through the inhibition of phosphorylation of AKT that induce a reduction in the”

RESPONSE: This issue was solved.

Line 667-668:  “The blockaded of TNF-R1 signaling might promote the proliferation of cells in the SVZ and neuroblast formation were enhanced [140]”

RESPONSE: This issue was solved.

Line 708: “that the adult neurogenesis in the amygdala response to hormones.”

RESPONSE: This issue was solved.

Line 1184: “In the striatum have been demonstrated”

RESPONSE: This issue was solved.

and etc.

RESPONSE: This issue was solved. An academic English editor service was hired to revise and proofread the manuscript.

Line 722: “Female C57BL6/J living 40 under environmental enrichment showed an increase in the expression of proteoglycan neuron-glia 2 (NG2) a marker that represents parenchymal precursor cells.” The citation is missing.

RESPONSE: This issue was solved. The citation was added.